# A Review of Epidermal Flexible Pressure Sensing Arrays

**DOI:** 10.3390/bios13060656

**Published:** 2023-06-15

**Authors:** Xueli Nan, Zhikuan Xu, Xinxin Cao, Jinjin Hao, Xin Wang, Qikai Duan, Guirong Wu, Liangwei Hu, Yunlong Zhao, Zekun Yang, Libo Gao

**Affiliations:** 1School of Automation and Software Engineering, Shanxi University, Taiyuan 030006, China; 202222207041@email.sxu.edu.cn (Z.X.); 202222207001@email.sxu.edu.cn (X.C.); 202222207011@email.sxu.edu.cn (J.H.); 202123601010@email.sxu.edu.cn (X.W.); 202023603007@email.sxu.edu.cn (Q.D.); 2School of Biomedical Engineering, Shanghai Jiao Tong University, Shanghai 200030, China; 3Pen-Tung Sah Institute of Micro-Nano Science and Technology, Xiamen University, Xiamen 361102, China; wuguirong@stu.xmu.edu.cn (G.W.); huliangwei@stu.xmu.edu.cn (L.H.); zhaoyunlong@stu.xmu.edu.cn (Y.Z.); 4Discipline of Intelligent Instrument and Equipment, Xiamen University, Xiamen 361102, China; 5Key Laboratory of Instrumentation Science and Dynamic Measurement Ministry of Education, North University of China, Taiyuan 030051, China; b20210626@st.nuc.edu.cn; 6Innovation Laboratory for Sciences and Technologies of Energy Materials of Fujian Province (IKKEM), Xiamen 361005, China

**Keywords:** epidermal sensing arrays, materials, structures, fabrication processes, back-end circuits

## Abstract

In recent years, flexible pressure sensing arrays applied in medical monitoring, human-machine interaction, and the Internet of Things have received a lot of attention for their excellent performance. Epidermal sensing arrays can enable the sensing of physiological information, pressure, and other information such as haptics, providing new avenues for the development of wearable devices. This paper reviews the recent research progress on epidermal flexible pressure sensing arrays. Firstly, the fantastic performance materials currently used to prepare flexible pressure sensing arrays are outlined in terms of substrate layer, electrode layer, and sensitive layer. In addition, the general fabrication processes of the materials are summarized, including three-dimensional (3D) printing, screen printing, and laser engraving. Subsequently, the electrode layer structures and sensitive layer microstructures used to further improve the performance design of sensing arrays are discussed based on the limitations of the materials. Furthermore, we present recent advances in the application of fantastic-performance epidermal flexible pressure sensing arrays and their integration with back-end circuits. Finally, the potential challenges and development prospects of flexible pressure sensing arrays are discussed in a comprehensive manner.

## 1. Introduction

Human skin can perceive external stimuli and form a subtle interaction with the external environment, which not only serves as a natural barrier against the outside world but also helps us respond to various external stimuli [1]. In recent years, with the development of Internet of Things (IoT) technology, electronic skin [2,3], human-machine interaction systems [4], and wearable devices [5] have attracted more and more attention, and people are committed to developing flexible wearable devices that can be comparable to the excellent functions of human skin. To meet the requirements, a lot of exploration and innovation in sensing array materials [6,7,8,9], microstructures [10,11], and fabrication processes [12,13] have been carried out to develop fantastic performance sensing arrays for practical applications. We focus on all epidermal pressure sensing arrays with excellent sensing properties from both material and structural aspects. The materials used for flexible wearable pressure sensing arrays are generally classified as carbon-based materials [14,15,16], metallic materials [17,18], polymeric materials [19,20], and some new materials [21,22]. Polymers such as polydimethylsiloxane (PDMS), polyethylene terephthalate (PET), polyimide (PI), Ecoflex, and polyurethane (PU) that have excellent biocompatibility, a low Young’s modulus, and ductility [23,24] would allow sensing arrays to better fit the skin for the purpose of collecting weak physiological signals. Liquid metal, carbon-based materials, and some new materials such as poly(3,4-ethylenedioythiophene):poly(styrenesulfonate) (PEDOT:PSS) possess fantastic electrical conductivity as electrode materials [7], and various electrode material systems are described in order to help researchers develop electrodes with more excellent properties. The sensitive layer is the core component of the pressure sensing array, so the materials used to make the array need to have excellent pressure-sensitive properties. Common materials for sensitive layers include hydrogels and conductive polymers. Polyvinylidene fluoride (PVDF) and its copolymer PVDF-trifluoroethylene (PVDF-TrFE) are used for dynamic multimodal testing of sensors [25,26] and insulation of printed short-circuit plates due to multiple excellent properties such as elasticity, low thermal conductivity, fantastic chemical resistance, and heat resistance [27]. Maita et al. [28] proposed an ultra-flexible tactile sensor with an embedded readout circuit and a PVDF-TrFE capacitor for ultra-flexible tactile sensors. In addition to the material itself, the fabrication process of the material also has a great influence on the performance of the sensing array, and the current common fabrication processes include 3D printing [29,30], screen printing [31], and laser engraving [13]. However, simply exploring different materials as sensitive elements for the fabrication of flexible sensing arrays does not result in sensing arrays with extremely superior sensing performance. For example, most polymer elastomers reduce the response speed of sensing arrays due to their viscoelasticity and adhesion [10,32], and the response time responds to the speed of signal acquisition and processing. In practical applications, especially in real-time monitoring of patient physiological information, slow response times can cause failure to reflect the patient’s physical condition quickly and accurately, resulting in untimely patient treatment. To solve these problems, researchers have designed different microstructures in the layer structures of sensing arrays to obtain a larger elastomeric strain and a larger internal contact area, which can improve sensing sensitivity, detection range, response time, and other performance indicators [33,34,35]. Guo et al. [36] microstructured the sensitive layer of sensing arrays, and compared to nonstructured sensing arrays, the sensitivity improved by tens of times. In addition, the encapsulation of sensing arrays is also a crucial part, and the encapsulation not only protects the devices at the physical level but also more critically prevents the erosive effects of water, oxygen, and corrosive liquids in the external environment [37,38]. The selection of suitable encapsulation materials, encapsulation forms, and encapsulation methods can further improve the reliability and stability of sensing arrays.

There have been a large number of researchers who have reviewed and discussed flexible pressure sensors. However, there have been few reviews discussing flexible pressure sensing arrays. While sensors can only display the collected physiological signals through simple electrical signals and graphical changes, sensing arrays can visualize the sensed pressure and change the perception of pressure from an abstraction to an image. In this paper, we will discuss the pressure sensing array back-end circuitry. Back-end circuitry, which is used to process the signal from the sensing front-end, is an important part of sensing technology. When a sensing device works, the sensing front-end is disturbed by the external environment (temperature, humidity) and some complex factors (e.g., electrocardiogram (ECG) signals can be disturbed by human motion), and abnormal signal peaks appear. These abnormal signals are mixed with normal signals, and the device can generate false alarm problems. However, the processing of the back-end circuitry (filtering, amplification, and digital-to-analog conversion) will make the output information of the sensing array much more credible. Later in the article, we will specifically discuss the application of sensing array back-end circuitry in filtering, amplification, and digitization techniques. When preparing flexible wearable electronic devices, optimization of the device is considered. For example, reducing the device’s size can make it more lightweight. However, the smaller the device size is, the higher the power density and thermal density of the device will be, which will have an impact on the electrical and thermal characteristics of the device. To solve this problem, new materials, new structures, and other methods can be used to optimize device performance and ensure the reliability and stability of the device. Reducing the power consumption of the device can extend the battery life and make the device more practical by optimizing the circuit design and reducing the power consumption in the circuit, such as by using low-power integrated circuits and reducing the circuit impedance. By optimizing the software design and reducing the use of the central processing unit (CPU), memory, and other devices, such as by using more efficient algorithms, reducing data transmission, etc., improving the response speed of the system can improve the interactive experience of the device and increase user satisfaction. By optimizing hardware design, such as using more efficient integrated circuits and increasing the operating frequency of devices, the system’s operation speed can be improved, and by optimizing software design, such as using asynchronous programming and reducing data transmission, the system’s response speed can be improved. The manufacturing cost of sensing arrays can be reduced by simple machining processes or by using common materials. In addition, we must also consider the contact between the sensing array and the skin at the front end of the device, including sweat, skin adhesion, and device breathability. Biodegradable sensing arrays made from paper-based substrates are currently receiving a lot of attention due to their extraordinary breathability, eco-friendly properties, and simple processing techniques.

In this paper, we will discuss the materials, structures, latest applications, and integration aspects of the back-end circuitry of the epidermal pressure sensing array, as shown in Figure 1. First, we summarize the materials and fabrication processes commonly used to prepare pressure sensing arrays and analyze the advantages and disadvantages of various materials and fabrication processes. An overview of the microstructures of the sensing arrays and their various applications on the skin surface is presented in Section 3 and Section 4. Subsequently, the latest research on the integration of sensing arrays with back-end circuits is outlined in Section 5. Finally, we discuss the future challenges and trends of pressure sensing arrays. We hope that this review will improve a comprehensive and systematic understanding for future researchers and provide them with ideas for the future design and development of pressure sensing arrays.

## 2. Materials Overview

### 2.1. Different Materials

Flexible pressure sensing arrays have been developing more rapidly in recent years. At the same time, polymers, hydrogels, metals, and carbon-based materials have received a lot of attention as flexible materials. Common polymeric flexible materials include PDMS, PET, PI, PU, and polyethylene naphthalene (PEN) [39], which are widely used as substrates for flexible sensing arrays due to their respective excellent physical and chemical properties. Hydrogels have excellent mechanical properties, extraordinary biocompatibility, and can adhere well to human skin [40]. By combining with other conductive materials, hydrogels have excellent electrical conductivity [41]. Therefore, hydrogels are also the main material for manufacturing flexible sensing arrays. Metal- and carbon-based materials are also widely used in flexible sensing arrays because of their extraordinary flexibility, fantastic electrical conductivity, biocompatibility, and a series of other advantages. These functional materials determine the overall performance of the sensor [24]. In this section, flexible materials are introduced in three main aspects: substrate materials, electrode materials, and sensitive layer materials. 

#### 2.1.1. Substrate Materials

The substrate material of the flexible sensing arrays plays an important role in the manufacturing of the sensing arrays, and the selection of a suitable substrate material can help the sensing arrays obtain better performance. The epidermal flexible pressure sensing arrays are in direct contact with human skin, so they must have extraordinary ductility to make them fit the skin more perfectly [40]. 

PDMS has extraordinary flexibility and chemical stability [24] and is often used to make flexible pressure sensing arrays with fantastic sensitivity. In addition, PDMS has excellent mechanical properties with a Young’s modulus of 2.2±0.2 MPa [42], which is shown in Figure 2 in comparison with the skin modulus of elasticity [43]. The fracture strength can reach 1–6 MPa and can still recover its initial state in general when stretched up to 200–500%. The biggest advantage of PDMS over other substrate materials is that its structure can be somewhat modified according to the actual application [44]. PET is a milky white or yellowish polymer that has extraordinary folding resistance and can continue to work normally at high temperatures of about 60 degrees Celsius and low temperatures of −70 degrees Celsius, and its mechanical properties are hardly affected by high or low temperatures. PET has a Young’s modulus of 2.8–3.5 GPa and a fracture strength usually between 50 and 100 MPa with extraordinary tensile and flexural strengths. At the same time, PET has extraordinary stretchability and can generally recover to its original length at a stretch of 30–80%. PET material is also extremely cost-effective, so a large number of sensing arrays are chosen as their substrate. PI is widely used in various fields as a special engineering material. It has extremely extraordinary mechanical properties and can also work stably in long-term high- and low-temperature environments [45]. The Young’s modulus of PI is 1.5–3.0 GPa, and the fracture strength is usually 100–170 MPa, showing extraordinary tensile and flexural strength. However, the stretchability of PI is poor, and its elongation at break is only about 3–15%. More and more researchers are now considering the use of paper or textiles as substrate materials for pressure sensing arrays compared to traditional polymers. Compared to traditional polymers, paper is low-cost, biodegradable, and reusable [46], so paper-based flexible sensing arrays have their potential advantages. Textiles are also frequently used as substrates for everyday products because they have low cost and extraordinary permeability, and their rough surface provides special microstructures for sensors [39]. By combining textiles with conductive materials, flexible pressure sensors with excellent performance can be produced. Liu et al. [47] designed an all-textile pressure sensor with a fast response time due to the porous nature of the textile and its highly elastic structure and excellent mechanical properties, breathability, and washability because the sensor components are all textiles.

#### 2.1.2. Electrode Materials

Currently, flexible pressure sensing arrays are developing rapidly, and therefore the electrode materials used to make them are receiving a lot of attention. The collected signal to be processed is transmitted by electrodes to a processor and then presented as an electrical signal [40] to obtain the physiological information we want to know about the human body. This requires the electrodes of the sensing arrays to have both extraordinary flexibility and fantastic conductivity. Common electrode materials include metallic nanomaterials, carbon nanomaterials, and so on. Choosing the suitable electrode material from these will lead to a substantial improvement in the performance of the sensing arrays.

Common metallic nanomaterials include nanoparticles, nanowires, nanosheets, and so on. These metallic nanomaterials have extraordinary electrical conductivity and stable chemical properties and are widely used in electronic devices [48,49,50,51,52]. The shape characteristics of gold nanomaterials can be readily adjusted, and their non-toxic properties make them biocompatible. Although gold nanowires have lower electrical conductivity compared to silver and copper nanowires, they are biologically safe [18]. Peng et al. [53] designed a chitosan substrate (CS) and used gold nanofibers as an electrode material for this electronic skin, as shown in Figure 3A. This study demonstrated that gold nanofibers have high light transmission and strain-insensitive conductivity and are high-quality materials for the fabrication of flexible electronic devices. However, their high cost limits their development for sensor applications.

Liquid metals have also been frequently used in recent years as materials for preparing sensors. The addition of liquid metal to elastomers can lead to a significant improvement in their toughness. Lou et al. [21] reported a liquid metal-polyvinyl alcohol (LM-PVA) composite, as shown in Figure 3B. The LMPVA film was obtained by ultrasonically dispersing liquid gallium in ethanol, mixing the suspension with the PVA/water solution, and then drying the film. The strain applied to the film resulted in a decrease in the resistance of the film, with a sensitivity of 0.8, proving its high electrical conductivity. It is also proven through experiments that the film has an extraordinary ability to recognize pressure, which further indicates that it has an extraordinary development prospect in the field of pressure sensors. In addition, the biocompatibility of gallium-based generics is also an issue we need to focus on. It was demonstrated experimentally that the toxicological monitoring of liquid metal nanoparticles (LMNPs) in female mice injected with dose 45 $mg\;{kg}^{-1}$ for three months showed normal important liver function markers, indicating that eutectic gallium indium nanoparticles were not intrinsically toxic and other important physiological parameters did not show abnormalities. Meanwhile, there was no significant upregulation of immunoglobulin E in the mice, indicating no signs of allergy and further demonstrating the excellent biocompatibility of the gallium-based alloy [59].

Conventional single-metal materials, in most cases, cannot meet our requirements for sensor performance, which requires us to prepare composite materials that combine the properties of multiple metals at the same time. Choi et al. [54] designed a wearable electronic device based on silver-gold nanocomposites, as shown in Figure 3C. It was demonstrated that the inert gold shell effectively prevented the oxidation of silver nanowires and the leakage of silver ions while retaining the fantastic electrical conductivity of silver nanowires, which effectively improved the performance of the electronic device. In future research, we should study more combinations of various materials to fully exploit the unique advantages of each material.

Carbon nanomaterials occupy an absolutely important position in the field of sensors because of their flexibility, fantastic electrical conductivity, fantastic specific surface area, low cost, extraordinary chemical stability, and simple fabrication process. In nature, carbon resources are abundant, and therefore carbon nanomaterials are well suited for large-scale production activities [40].

Carbon nanotubes (CNTs) have the advantages of extraordinary electrical conductivity [60], low density, and high aspect ratio [23], and their unique one-dimensional structure makes them possess strong physical properties [61,62], which can effectively improve the performance of sensors, thus becoming the most widely used one-dimensional carbon nanomaterials, which are widely used in flexible wearable sensors. Graphene, a two-dimensional carbon nanomaterial, is one of the most important materials for making flexible pressure sensors because of its excellent electrical conductivity, strong mechanical properties, easily adjustable chemical composition, unique electronic structures, low cost, and easy processing and fabrication [63,64,65]. Two-dimensional graphene is often used as an important material for preparing sensors because of its fantastic flexibility and mechanical strength. Shin et al. [66] used a silver nanowire-graphene hybrid transparent electrode in addition to a transparent flexible pressure sensor. It was demonstrated that the entire sensor has 83% light transmission at a wavelength of 550 nm, the sensor can still work properly when the pressure reaches 3 MPa, and the response time of the sensor is very short. This indicates that the pressure sensor has fantastic transmittance, fantastic conductivity, and fantastic sensitivity. Therefore, transparent pressure sensors based on silver nanowires and graphene are promising for electronic skin applications.

MXene, as a novel two-dimensional carbon nanofiber material, has the advantages of fantastic electrical conductivity, unique structures, a high specific surface area, strong hydrophilicity, and extraordinary mechanical properties [64,67,68,69]. Due to these excellent properties, MXene is expected to be one of the main materials for the fabrication of flexible pressure sensors in the future. The extraordinary electrical conductivity and its unique structures allow MXene to have a large range of resistance adjustment and to respond quickly to the applied pressure. The high specific surface area facilitates its tight bonding with other materials, resulting in composites with even better properties [70]. Most MXene compositions are composed of carbon (C), nitrogen (N), and the non-toxic transition metals titanium (Ti), niobium (Nb), and tantalum (Ta). The MXene nanosheets showed extraordinary biocompatibility by demonstrating essentially no effect on various vital parameters and no abnormal behavioral activity in mice [71,72]. However, MXene is easily oxidized in a humid environment, and the conductivity and structure of the oxidized MXene nanosheets may change unpredictably. Ma et al. [55] designed a flexible pressure sensor based on the controlled oxidation of MXene, and the schematic diagram is shown in Figure 3D. By investigating the changes in the conductivity of MXene nanosheets at different oxidation levels, the oxidation level of MXene that makes the sensor sensitive is found to be the highest. In addition, it was found that the sensitivity of the MXene pressure sensor based on commercial A4 paper is higher than that of the MXene pressure sensor based on filter paper because the A4 paper has wrinkled microstructures and some fine fibers compared with the filter paper, which will make its contact resistance change in a wider range and thus improve its sensitivity.

Daily carbon ink (DCI) has a very low cost compared to other carbon materials. In addition, by applying DCI to paper, we can clearly measure its resistance, which is less than 0.17 m, proving its excellent electrical conductivity [73]. Duan et al. [74] prepared a paper-based pressure sensor using paper and DCI. In this sensor, carbon nanoparticles filled the small holes on the paper surface, maintaining a continuous conductive path through the paper. The sensitivity can reach 0.614 kPa^−1^ when the pressure is in the interval 0–6 kPa, and in the interval 6–40 kPa, the sensitivity is 0.064 kPa^−1^. In addition, the sensor was cycled under 2 kPa pressure conditions, and the experimental results showed that after 4300 cycles, the current response of the sensor only decayed by 23%, which basically met the usage requirements. Therefore, the DCI paper-based pressure sensor can be widely used in human health monitoring.

Lo et al. [22] reported a stretchable polymer based on PEDOT:PSS. The PEDOT:PSS/poly(ethylene oxide) (PEDOT:PSS/PEO) polymer film was prepared by mixing PEO with the PEDOT:PSS solution. By blending PEO and ethylene glycol (EG) solvent with PEDOT:PSS/PEO, the brittleness of PEDOT:PSS/PEO was eliminated, resulting in a significant improvement of its mechanical properties, with its tensile properties reaching 50%. The film also possesses a bottom sheet resistance of 84 Ω, which was used as a printed electrode to prepare a sensor capable of effective monitoring of pulse and heartbeat. Keum et al. [56] reported a high-K ionic gel film, which was prepared schematically in Figure 3E. The sensitivity of the sensor prepared from this film as a sensing element reached 146.2 $nF\;kPa^{-1}$ (0–1.3 kPa) and 308.5 nF kPa^−1^ (1.3–4 kPa). The response times for loading and unloading were 272 ms and 234 ms respectively when 2 kPa was applied externally and the frequency was 100 kHz and they increased to 670 ms for both loading and unloading when the frequency was reduced to 100 Hz. Although the response time is relatively long compared to other pressure sensors, it is fully adequate for monitoring human physiological activities. Flexible polyester fiber conductive tape, a new industrial product, is also often used as an electrode material for flexible pressure sensors. Flexible polyester fiber conductive tape is not only flexible, but its surface roughness also greatly enhances the sensing performance of flexible pressure sensors [75]. Duan et al. [76] prepared a paper-based pressure sensor using polyester fiber conductive tape as an electrode. The sensitivity of the sensor was reached at 5.54 kPa^−1^ when the sensor was in the pressure range interval 0.5–5 kPa, and when the pressure was 5–60 kPa at 1.61 kPa^−1^. Compared with the traditional copper electrode, polyester fiber conductive tape as the electrode and paper contact can form more contact points, produce microstructure deformation, and cause changes in the current. While polyester fiber conductive tape is a low-cost material, the flexible pressure sensor based on polyester fiber conductive tape has fantastic prospects for development in the field of wearables.

This section introduces the electrode materials through metal and carbon-based nanomaterials and some new materials as sensor electrode materials, which have different dimensions, and the different dimensions lead them to have different structures and properties, and their properties are compared as shown in Table 1. By combining materials of different dimensions according to certain physicochemical properties, we can obtain multifunctional and flexible sensors. This provides certain directions for future research. For example, the combination of a one-dimensional structure and a two-dimensional structure allows the composite material to have excellent flexibility, enabling it to have fantastic mechanical properties.

#### 2.1.3. Sensitive Layer Materials

As the core component of a flexible pressure sensor, the sensitive layer material has a great impact on the performance of the sensing arrays. In order to achieve detection and quantification of contact pressure, it is necessary to convert pressure information into electrical signals. Similar to pressure sensors, pressure sensing arrays can be classified into four types: piezoresistive, capacitive, piezoelectric, and triboelectric [85]. Each sensor has unique characteristics due to the differences in active materials and device structures. Piezoresistive pressure sensing arrays measure pressure based on the change in electrical resistance, which changes when pressure is applied to the sensing array. Carbon-based materials are the most common materials used as sensitive layers for piezoresistive sensing arrays. Three-dimensional foams of graphene or carbon nanotubes can be synthesized by processes such as chemical vapor deposition (CVD), freeze-drying reduction, blade coating, vacuum filtration, laser scribing, and the addition of vitamin C. Carbon fiber structures can be obtained from carbonized polymer nanofibers and silk fibers. Carbonaceous nanomaterial films can be laminated on the surface of substrates with regular or irregular microstructures by spraying or transfer processes to form microstructure-sensitive layers. Different materials form from structures that undergo different degrees of deformation due to stress concentration, thus improving pressure sensitivity. Moreover, Mxene and metallic materials (nanoparticles, nanowires) are also commonly used as sensitive layers for piezoresistive sensing arrays. Piezoresistive sensing arrays are the most commonly used pressure sensing arrays because of their simple device structure, relatively wide operating range, low operating voltage, low energy consumption, and easy processing and fabrication. However, piezoresistive pressure sensing arrays have the problems of slow readout speeds and significant crosstalk [86]. A capacitive pressure sensing array usually consists of two parallel electrodes and a dielectric layer. The variation of the capacitance value depends on the electrode spacing, the effective contact area, and the dielectric constants of the different dielectrics. The capacitance value changes when pressure is applied to the sensing array. Sliver nanowires (AgNWs) are widely used in the sensitive layer of capacitive pressure sensing arrays because of their excellent electrical conductivity and adaptability to different microstructures. Or they are doped with polymers to form dielectric layers of composite nanomaterials. In addition, carbonaceous nanomaterial foams are often used as dielectric layers for capacitive sensing arrays. Compression of the dielectric layer using different materials and structures leads to changes in the interfacial polarization and an increase in the relative dielectric constant, which leads to an increase in the sensitivity of the sensing array. The main advantages of capacitive sensing arrays are stable sensing performance, fantastic dynamic response, and low power consumption. However, due to the viscoelasticity of the polymer dielectric layer, capacitive sensing arrays can suffer from problems such as slow response [48]. The piezoelectric pressure sensing array is based on the piezoelectric effect to measure pressure, and when pressure is applied to the sensing array, its internal polarization phenomenon occurs while positive and negative charges appear on its two opposite surfaces. For piezoelectric pressure sensing arrays, the introduction of carbonaceous nanomaterials into the piezoelectric material can improve the pressure sensitivity. In addition, to further improve the piezoelectric output of piezoelectric materials, researchers have developed various flexible piezoelectric pressure sensing arrays composed of metal nanoparticles and PVDF composites. Piezoelectric sensing arrays respond to external pressure by generating transient electrical signals and have the advantages of fast response and extraordinary sensitivity. In addition, unlike piezoresistive and capacitive types, pressure sensing arrays based on the piezoelectric effect are self-powered. However, because the output voltage of the piezoelectric sensing array and the current generated by the external circuit are pulsed, they are not suitable for measuring static pressure [87]. Triboelectric pressure sensing arrays are based on the triboelectric effect to measure pressure. When pressure is applied to the sensing array, materials with different triboelectric properties accumulate different charges on the surface when they come into contact. Carbon nanotubes are commonly used as a friction layer for triboelectric nanogenerators due to their surface roughness. Similarly, nanofibers doped with AgNWs have a rough surface and therefore have a higher triboelectric output. Triboelectric sensing arrays are also self-powered sensing arrays, and the difference in polarity of the triboelectric layer material has a significant impact on the output performance. Triboelectric nanogenerators (TENGs) are considered the most promising candidates for pressure sensing arrays due to their extraordinary power output, low cost, and simple preparation process [87]. 

Meanwhile, pressure sensing arrays require a sensitive layer material that is very sensitive to pressure and can achieve a wide pressure sensing range. A strain sensor based on a conductive polymer hydrogel was reported by Shen et al. [57], and the process of preparing the material is shown in Figure 3F. The hydrogel itself has a structural domain that allows the sensor to exhibit extraordinary stretchability and a wide detection range. The study also demonstrates that the excellent mechanical properties inherited from the hydrogel allow the sensor to have excellent stability and durability. The sensing performance and hysteresis of this sensor did not change significantly after 2000 cycles of cyclic hysteresis during loading and unloading under 100% extraordinary strain conditions. Some conventional hydrogels may have insufficient mechanical properties, which to some extent limits their application in the field of flexible electronics. Zhou et al. [58] prepared a new type of hydrogel by mixing hollow polyaniline spheres (HPS), PVA, and phytic acid (PA) in a solution, and the fabrication process is shown in Figure 3G. This new hydrogel effectively improved the mechanical properties of the material, and the pressure sensor prepared by adding a dielectric layer between this hydrogel exhibited extraordinary sensitivity, making this material a potential material for making pressure sensors. Xu et al. [88] designed a pressure sensor based on conductive polyaniline (PANI). In this experiment, the volatility possessed by polyaniline monomer was utilized, and layers of conductive polyaniline grew vertically along the gas-liquid interface of electrostatically spun fibers. It was demonstrated that PANI can effectively improve the efficiency of electron transfer on the conductive fiber, while the fiber membrane has a high specific surface area, providing a fantastic structural basis. The extraordinary sensitivity, wide detection range, and fast response time of the sensor make the material promising for wearable sensor applications. Duan et al. [89] prepared a low-cost (only $0.007 (RMB 0.05) for a 5-cm length sensor) and environmentally friendly strain sensor based on DCI and elastic corespun yarn. In this sensor, a large number of carbon nanoparticles are stably adsorbed on the elastic core yarn (coarse polyester fiber), which effectively improves the strain sensing response. The sensor exhibits a fantastic linear response (*R*^2^ = 0.98100) in the strain range of 0.5–20%, and the strain sensitivity is about 6.1. Therefore, the sensor is widely used for motion detection in different parts of the human body.

### 2.2. Material Fabrication Process

#### 2.2.1. 3D Printing

3D printing, also known as additive manufacturing, was first implemented in the 1980s and has received increasing attention from researchers over the past 30 years. 3D printing is a technology that uses a digital model as the basis for manufacturing the desired object by accumulating layers of materials from the bottom up. By manufacturing products through this technology, the manufacturing difficulty of the product as well as its production cost can be effectively reduced [90]. Compared to the traditional manufacturing process, 3D printing is able to experiment well with the fusion of materials of different dimensions, control the thickness of electrodes flexibly, prepare devices with superior performance, and have a fantastic environmental value. At present, the rapidly developing 3D printing technology has been used in various engineering fields, such as architecture, electronics, aerospace, and so on. There are already relevant researchers who have started to use 3D printing technology to fabricate flexible sensors [91,92].

3D printing, as a digital technology, is very different from traditional manufacturing methods. It is based on the principle of converting a digital model file into the desired solid model by means of a digital material printer. It generally consists of four steps: the first step is that the structural model of the product is constructed in the computer; the second step is that a standard format file is generated; the third step is that the data of the model is layered; and the fourth step is that a suitable printing material and printer are selected to start the printing process [93]. The schematic diagram is shown in Figure 4A. The main 3D printing technologies we commonly see today are light-cured 3D printing, fused deposition modeling (FDM), inkjet printing (IJP), and direct ink writing (DIW) based on material extrusion, among others. Details are shown in Table 2.

Light-cured 3D printing based on light-curing enables the rapid fabrication of polymers with high resolution, along with a small footprint and low cost compared to other 3D printing methods [102]. Based on these advantages, various objects prepared by light-cured 3D can be customized as a way to adapt to different situations. Light-curing-based 3D printing technologies can be classified into several different types depending on the different principles of pattern formation and the control system, with the common ones being digital light processing (DLP), stereolithography appearance (SLA), and liquid crystal display (LCD) [29]. DLP 3D printing uses digital micromirror elements in the production process and then projects the product cross-section onto the surface of liquid photo-resin, which causes the photo-resin to be light-cured layer by layer. SLA 3D printing technology is the earliest 3D printing technology, so its application in various fields is more mature. Its working principle is to use the scanning head to scan the laser beam to the surface of the resin, let it be exposed, cure the part of the liquid resin that is illuminated by the laser, completing the processing of the cross-section, and then use the same method to irradiate and cure a new layer of cross-sectional profile at that level. Repeating the above operations so that the layers of cross-sectional profile are gradually laminated together to form the final product. Based on its working principle, SLA technology has certain advantages in the production of large-size products. However, SLA technology has disadvantages such as the need to design support structures in advance, the high cost of equipment, allergies to human skin, and a slow molding speed, which limit the development of the technology to a certain extent. LCD 3D printing technology uses liquid crystal displays as the imaging system, and liquid crystal displays have a high resolution. Wu et al. [103] successfully prepared a highly flexible and self-healing hydrogel material for the fabrication of flexible wearable sensors by LCD 3D printing. This study demonstrated that light-cured 3D printing can be used to customize small parts to assemble objects with complex structures that are likely to meet our requirements. However, its high fabrication cost and the large computational effort of preprocessing are limiting its development to some extent [104].

FDM technology is a nozzle-based deposition system that prints the desired model by depositing molten material in layers on a heated plate through extrusion of the nozzle based on a product structure constructed on a computer [105]. The material used in FDM technology is usually a thermoplastic polymer, mostly in the form of filamentary wire. The printing process is roughly as follows: First, a thermoplastic filament is placed in the printer. When the nozzle reaches the desired temperature, the filament is fed into the extrusion head and melted in the nozzle. The extrusion head is connected to a three-axis system that allows it to move in both X and Y directions. The melted material is extruded into thin strips and deposited layer by layer at predetermined locations, where it cools and cures. Finally, when one layer is printed, the table drops in thickness by one layer in predetermined increments along the Z-axis direction, and the material is extruded and deposited on top of the previous layer of cured material, forming the final product by stacking the material layer by layer. FDM technology does not require the use of lasers, so the equipment is simple to operate, has a low cost, has a high utilization of raw materials, and does not produce pollution, which has a minimal impact on the environment. Davoodi et al. [106] used the FDM 3D printing technique to prepare strain sensors. The process uses safe materials and allows fantastic control of the internal pore structure of the material to achieve controlled sensing performance. The sensor can maintain a stable piezoresistive signal even after repeated experiments and has fantastic flexibility and biocompatibility, making it a promising development in the field of wearable sensors. However, it has certain drawbacks. Compared with other printing technologies, FDM has lower dimensional accuracy and resolution, which is not suitable for preparing precision parts, and the surface of the prepared objects is relatively rough and requires additional processing.

DIW technology, based on material extrusion, involves the pressure-driven deposition of viscoelastic inks through fine nozzles. This is due to the rheological properties of the ink used, which allow it to be extruded from the fine nozzle at room temperature and deposited on the construction table under external pressure to create the desired object. It has attracted more and more researchers because of its low cost, ease of operation, and ability to bond different materials. Unlike other printing technologies, DIW does not require high material requirements, and multi-material structures can be achieved through its single-step processing, a property that saves manufacturing costs while retaining the excellent properties of the material [100]. Wang et al. [107] investigated a high operating temperature DIW (HOT-DIW)-based 3D printing system to fabricate soft silicones. Extraordinary resolution soft silicon of 50±5 μm can be printed at 45 °C using a wax particle (PSW) phase change ink through a 50 μm nozzle. The study demonstrated that the functionalization of this 3D printing system and the ease of printing with multiple materials greatly enhanced the performance of flexible electronics. DIW can also contribute to the production of flexible electronics by creating a material structure that improves the sensitivity of the pressure sensor; by adjusting the concentration of nanomaterials in the ink, the contact resistance of the sensor can be changed, thus affecting the sensitivity of the sensor; and DIW can also be printed with conductive nanomaterials, which can make the sensor more conductive. This can make the sensor more conductive. However, low resolution and nozzle clogging may occur during the process, which needs to be further optimized in future research.

#### 2.2.2. Ink Jet Printing

IJP technology was initially used for pattern printing in two-dimensional space, and with the development of technology, it was gradually extended to 3D ink jet printing [80], using a layer-by-layer printing method that enables the construction of materials in the vertical direction. Ink jet printing technology allows the simultaneous application of different materials to the substrate, which is a particular advantage of ink jet printing [81]. IJP is a rapid prototyping technology that works by converting computer-aided design (CAD) design model data into path control information, controlling the movement of the print head and ink jet volume through the printer control system, and forming the final product by stacking ink layer by layer. Compared to traditional inkjet printing, IJP has an optimal resolution of 2 μm [108], and it can produce more complex products with a faster curing speed. When using IJP as a sensor preparation process, there are three key issues to consider: The first is the choice of ink, ink viscosity, and surface tension. The design of inkjet printing inks needs to focus on the characteristics of the droplets so that the base material can be better bonded together; another is the ink adhesion to the selected substrate. Improved adhesion can be achieved by plasma treatment of the surface of the base material to make it hydrophilic, so the ink particles can adhere well to its surface; or ink sintering, where a high sintering temperature can dissolve additional solvents to allow further integration of nanoparticles to improve the conductivity of the sensor, but also to a certain extent will damage the performance of the base material; finally, the need to consider the mechanical durability, based on the preparation of the sensor, needs to be in the sensor for a long time. Based on its preparation, the sensor needs to be under pressure for a long time and be able to maintain its sensing performance [108]. Ma et al. [109] prepared a double-layer flexible pressure sensor using inkjet printing technology. The micro-concave-structured stencil of the sensor was prepared from polyacrylic acid (PAA) suspension as an ink. The sensor has excellent performance, with a sensitivity of up to 3900 kPa^−1^ when 0–1 kPa. Furthermore, the sensor maintained excellent sensing performance after 1000 cycles of pressure loading and unloading in 1 kPa. The sensor was fixed on a human joint and monitored in real time, and the results showed that the sensor has excellent stability, further proving its promising development in the field of wearable devices. However, inkjet technology still has some limitations. First, the materials used are restrictive, and some materials cannot maintain stability at high temperatures and pressures [81]; second, although the printing accuracy has been greatly improved, it is still not comparable to the traditional process; finally, IJP technology requires some subsequent processing, such as polishing and cleaning, which requires additional processing time.

#### 2.2.3. Screen Printing

Screen printing technology refers to printing a complete two-dimensional pattern by printing ink on the substrate through a screen printing plate. The principle of screen printing is shown in Figure 4B. During printing, ink is poured on the upper surface of the printing plate, and a squeegee is used to apply pressure to one end of the screen printing plate while moving to the other end. The ink is squeezed onto the substrate during the movement, and there is a certain gap between the screen plate and the substrate during the printing process, which ensures the dimensional accuracy of screen printing. Screen printing as a material fabrication process has its own unique advantages. For example, compared to other printing techniques, screen printing is not limited by the substrate, and it is less costly and easier to operate [94,95,96,110]. Kim et al. [111] designed a sensor based on graphene and screen printing technology and confirmed the sensor has extraordinary sensitivity (53.1 ± 2.7 kPa), excellent conductivity (1.49 × 10^4^
s m^−1^), excellent chemical stability (the electric potential signal and sensitivity basically do not change under different pH values), and excellent durability (after up to 15 h of experiments, the sensor still maintains a stable electric potential). MXene, with its excellent performance, is also often used to prepare flexible devices. Wu et al. [112] prepared a piezoresistive sensor using MXene/xanthan gum as the ink using the screen printing technique. By testing the sensor under different pressure conditions, the current signal collected directly reflected the applied pressure, and the sensor was able to respond and recover quickly to the external pressure, which directly demonstrated the excellent sensing performance of the screen-printed sensor. Dong et al. [113] prepared an electrode array based on a liquid metal/polymer conductor using the screen printing technique. It was demonstrated that the array had extraordinary electrical conductivity, excellent flexibility, and excellent electrical properties even when subjected to 100% strain.

#### 2.2.4. Laser Engraving

Laser engraving technology is based on computer numerical control (CNC) technology, with the laser as its processing medium. The processing of the material by laser irradiation produces the physical properties of instantaneous melting and vaporization, thus completing the processing of the material. In recent years, optoelectronics has developed rapidly, and laser engraving technology has been applied to various fields, including flexible electronic devices [114]. The laser engraving machine is shown in Figure 4C. Through experiments, we learned that the response time of the sensor is less than 47.3 ms at 350 Pa, and after 3000 loading and unloading cycles, the load voltage maintains a stable periodic change, which further proves that the sensor has the advantages of fast response time and strong durability [13]. Wang et al. [115] designed a wearable pressure sensing system by laser engraving technique, which adjusts the microstructures of the sensor surface to achieve sensitivity control by laser engraving technique. It was also demonstrated that the sensor has the advantages of a fast response time and fantastic durability, which can be used for the detection of human physiological signals. Laser engraving technology can also significantly improve the working range of the sensor to some extent. Xu et al. [116] reported a silver-plated fabric using laser engraving as an electrode material for the sensor. This study used stainless steel as the engraved template and grafted a waterproof PDMS micropillar film with carbon nanotubes on its surface, which allowed it to work properly in extreme environments. Meanwhile, the conductivity of 0.05 $\Omega \;{cm}^{-2}$ was achieved by using a hierarchical structure prepared by laser engraving techniques. This study demonstrates that the electrode enables the sensor to operate over a range of 800 kPa while also combining extraordinary sensitivity, excellent durability, and a fast response time. Graphene is also often used as a flexible material. The properties of graphene can be enhanced by converting sp-3 carbon to sp-2 carbon through laser engraving. Chhetry et al. [117] used laser-engraved graphene to prepare a sensor with multifunctional features. The sensor was studied to have extraordinary sensitivity to strain while still functioning after extensive cycling tests, which is well suited to the needs of human electronic skin.

#### 2.2.5. Others

In addition to the three common fabrication processes mentioned above, there are also some fabrication processes that can effectively prepare pressure sensors, resulting in excellent structural and sensing properties. For example, electrostatic spinning [118], spraying methods [119], dry etching [120], freeze-drying [121], laminated [122], nanoimprint lithography (NIL) [123], and so on. Wang et al. [124] prepared layered nanofiber mats using electrostatic spinning, which was a water-assisted electrostatic spinning process, and by adjusting the parameters of electrostatic spinning, thus achieving control of the layered morphology and making it have more excellent properties. The drying process patterns the nanofiber mat, which has fantastic mechanical properties, high roughness, and durability and is able to detect the strength of the human pulse signal well, suggesting a new feasibility for wearable medical devices. Spray coating as a sensor fabrication process has the advantages of low cost and general simplicity. Yang et al. [119] prepared MXene-based interfinger electrodes by spraying and experimentally demonstrated that the sensor has a fast response time and recovery time. The sensor was found to be stable under repeated pressure cycles. At the same time, the sensor can clearly detect the fine signal (7.8 Pa), which is very helpful for human physiological signal acquisition and can be considered for electronic skin application. In addition, finding a large-area, low-cost processing technology could also facilitate the better development of flexible pressure sensors. Shen et al. [120] developed a lithography-free, mold-free fabrication process using an orbital etched film as a template, which allows the fabrication of conductive arrays over a large area, and the process is flexible enough to allow the sensor structures to meet our requirements by adjusting its various parameters. This study shows that the pressure sensor prepared by this method has a wide detection range ($<10^{-4}–>10^{2}$) and extraordinary sensitivity for both low and high pressure. Lamination is also a commonly used method to prepare sensors, and a novel single-layer sensor was prepared by Kim et al. [122] using lamination. The non-conductive wool yarn was impregnated with CNTs to obtain conductivity, and the sensing layer was obtained by weaving non-conductive fibers, conductive fibers, and electrodes, which were then thermally laminated so that both sides were covered with a flexible film. The sensor is very thin and has a fast response time, ensuring fast electrical performance in practical applications. In addition, after 20,000 cycles of pressure loading and unloading, the uniform resistance change is less than 7%, and after 50 ms cycles of washing and squeezing, the uniform resistance change is less than 10%, which further proves that the sensor has excellent durability and has fantastic potential for development in the sensor field. Chen et al. [123] prepared piezoelectrically enhanced vertically aligned PVDF-TrFE/BaTiO_3_ nanocomposite micropillar arrays using a nanoimprinting process. After using NIL, BaTiO_3_ nanoparticles were uniformly distributed in the nanocomposite micropillars. The piezoelectric coupling coefficients of PVDF-TrFE films and PVDF-TrFE/BaTiO_3_ composites were 14.6 and 35.3 ${pC\;N}^{-1}$, respectively. It was further demonstrated that the piezoelectricity of the prepared nanocomposites was enhanced, and the mechanical flexibility of the micropillar arrays was also effectively improved. When the flexible piezoelectric sensor prepared based on this array is used for the detection of human respiration, it can well distinguish different respiration signals such as deep breathing, gasping, strained breathing, and normal breathing, which can effectively detect the health condition of the human body and has been a fantastic development in human health detection. Therefore, it is very important to choose a suitable fabrication process for the processing of the sensor, which can improve the performance of the sensor under certain circumstances.

## 3. Array Structures Overview

### 3.1. Overview of the Overall Structure of the Array

In addition to the selection of sensing materials with excellent inherent properties described above, the overall layer structures of the array are a crucial factor in achieving extraordinary performance for the sensor devices. For sensing arrays, another major way to improve sensing performance is through structural design. Certain microstructures on the surface or inside the sensing element have been shown to be effective in improving the sensing performance of sensors in terms of pressure sensitivity, detection range, and response time. Scanning electron microscopy (SEM) can clearly characterize the morphology and dimensions of microstructures, while multiphysics field simulations can verify and explain the role of nano-microstructures [125]. This paper introduces various machinable micro-nanostructures, and in general, the sensing arrays can be broadly classified into two categories: bendable and stretchable structures. Based on this, the sensing arrays are finely classified according to the key layers that are usually treated as microstructures, and they are divided into electrode layers and sensitive layers, where the electrode layers include interdigital structure, sandwich structure, and single electrode structure. Sensitive layers include microgeometric structures (micro-pyramid, microneedle, micropillar, etc.), porous structures, multilayer structures, and hybrid multistructures. Next, we first describe the bendable structure and the stretchable structure according to the overall structures.

#### 3.1.1. Bendable Structure

Bendability has become an essential feature for wearable epidermal electronics to meet the development of flexible pressure sensing arrays for epidermal applications. The epidermal sensing array can continuously sense a large number of physiological parameters of the human body, human movement, and other indicators to monitor human health in real time. Pressure sensing is one of the most important functions of the human skin. In Gleskova et al. [126], it was noted that thin film transistors (TFT) fabricated on rigid or flexible substrates can reach 0.5% strain when bent under tension. However, at higher strains, the TFT on the rigid substrate delaminates, while 50% of the TFT on the flexible substrate still works. When compressed, TFT results are different on rigid and flexible substrates. Delamination occurs at 0.5% strain for the former and at least 2% strain for the latter. The mechanical failure during compression is determined by the bond between the substrate and the TFT structure. Therefore, having excellent bendability is a necessary prerequisite for sensing devices to achieve excellent performance while also conforming to the mechanical and bending stresses of human skin. There are different types, sizes, shapes, numbers, and distributions of sensory receptors on human skin, realizing different characteristics of pressure perception in different parts of the skin. For example, the skin on the fingers has different sensory fields and sensitivities than the skin on the body. Changes in physiological pressure may also reflect disease-induced deterioration of body tissues. Therefore, the property of compatibility of epidermal flexible pressure sensing arrays with human skin is particularly important, and bendable structures have subsequently received attention and gradually become the mainstream structure of flexible pressure sensing arrays [127,128,129], as shown in Figure 5A–C for a physical diagram demonstrating the bendability of flexible pressure sensing arrays. Although extraordinary sensitivity is an important sensing performance parameter, rigid pressure sensors are not bendable. In contrast, sensing arrays with a bendable structure can be better attached to a surface similar to human skin and are more practical. In contrast, sensing arrays with ultra-thin and ultra-bendable structures can be better attached to a surface similar to human skin and are more practical [130,131]. In conclusion, the flexible pressure sensing array with bendable structure is a wearable epidermal electronic device with wireless communication, is lightweight, and is compatible with human skin compared to traditional rigid sensing devices [132]. 

#### 3.1.2. Stretchable Structure

The flexible sensing materials discussed in Section 2 typically have fairly high conductivity and stretchability, with stretchability depending on the stretch limit of the substrate. However, the conductivity of the electrodes decreases dramatically under large pressure and strain, thus limiting the application of sensing arrays. The deposition of metal films on elastic substrates to form microcracks is one of the most common methods for preparing highly stretchable structural arrays. Other stretchable micro or macroscopic structures, such as fractal design and serpentine structure [134], coiled structure [135], out-of-plane wavy structure, open-mesh structure [136], kirigami structure [137], and three-dimensional porous structure [138], have also shown excellent stretchability. When developing epidermal flexible pressure sensing arrays, biocompatibility is always the first priority, and favorable stretchability of the arrays is the most important step. Therefore, stretchable structures, as one of the overall structures of sensing arrays (shown in Figure 5D), have a broad prospect for future applications in the development of wearable devices, human-machine interaction, and other fields [133,139].

### 3.2. Overview of Electrode Layer Structures

The electrode layer of the sensing array is one of the key parts of the overall sensing device that plays a role in sensing and measurement because the electrode layer deforms under pressure, which affects the resistance [65], capacitance [140], piezoelectric [141], and triboelectric [142] signal outputs. In addition to the innovative fusion of electrode layer materials to improve the performance of flexible pressure sensing arrays, the morphology and geometry of the electrode layers play a critical role in sensor performance. In this review, we focus on recent advances in electrode layer structures for sensing arrays and summarize the electrode layer structures into three major categories: interdigital structure, sandwich structure, and single electrode structure. To date, many researchers have proposed different electrode layer structures to improve the performance parameters of sensing arrays, such as sensitivity, detection range, and response or relaxation time. In the following, we will describe in detail the effects of these three major types of different electrode layer structures on various aspects of the sensing array’s performance.

#### 3.2.1. Interdigital Structure

The interdigital structure is a common electrode structure. Its structure mainly consists of two intersecting interdigital electrodes, which are isolated and fixed on a flexible substrate to form a tiny air gap, which can reduce the crosstalk problem of the sensing array to some extent [143]. In addition, the electrodes of the interdigital structure have a significantly larger number of electrodes and a larger surface area compared to individual electrodes, thus increasing the density of signal acquisition and thus the spatial resolution, which holds superb promise for sensing arrays applied to haptic perception [119,144]. Finally, the interdigital electrode structure can be prepared by micromachining technology with high processing accuracy and can be used repeatedly without frequent replacement, which saves cost. In summary, sensing arrays with interdigital structure have the advantages of small size, extraordinary sensitivity, fast response, high crosstalk resistance, and low cost and are widely developed [145], as shown in Figure 6A. In addition, Qin et al. [146] developed an extraordinary-sensitivity capacitive pressure sensor based on MXene/polyvinylpyrrolidone (MXene/PVP) and interdigital electrodes and integrated the sensing units into a 3 × 3 sensing array (as shown in Figure 6B) to achieve spatial information acquisition and pressure distribution mapping, where different numbers of coins and keys were placed on the surface of the array and different locations showed sensed voltages of different intensities. To obtain a sensor with better performance, Yang et al. [119] developed a method to enhance the sensor’s performance by combining an interdigital electrode as the electrode layer with a sensitive layer with microstructure treatment. A high-performance flexible pressure sensor was prepared by spraying MXene on a spiny PU substrate as the sensitive layer and an interdigital electrode as the electrode layer (as shown in Figure 6C). The sensor not only reflects its excellent performance in terms of sensitivity and stability but also highlights a certain self-healing function, and its sensing sensitivity can reach 85% of the original one after the layer structure is damaged. In addition, in order to explore its haptic feedback capability and its application in the field of artificial electronic skin, a 4 × 4 full-jet MXene piezoresistive sensing array is also integrated and designed, and its very promising application in robotics and electronic skin applications is demonstrated through experiments.

The electrode and sensitive layers of a sensor or sensing array are processed into different structures, which affect their performance. The sensing performance reflects a diversity of variations due to the arrangement within the different structures, the uniformity of their distribution, and their own differences. This is due to the fact that the deformation that occurs in different structures when pressure is applied is different, and thus the effect on the performance of the sensing device can vary. In conclusion, the use of flexible pressure sensors or sensing arrays with different structures is a common strategy to improve their sensitivity, widen their operating range, enhance their stability, and obtain excellent performance such as fast response time and low hysteresis time. Currently, researchers usually apply various different microstructures to one sensor or sensing array to achieve simultaneous optimization of multiple sensing performances or linear improvement of one sensing performance. Details are shown in Table 3.

#### 3.2.2. Sandwich Structure

The electrode layer of a flexible pressure sensing array generally consists of two electrodes sandwiched between a sensitive layer structure, so it is usually referred to as a “sandwich” structure. As mentioned in Section 2, a number of materials have been selected by different fabrication processes to enable excellent sensor sensing performance. However, it is worth noting that since the sensing performance of pressure sensing arrays is evaluated by many important parameters together, it is not possible to optimize all performance parameters together just by material selection, processing, and doping variations. Therefore, we can perform a series of processes on the structure of the sensing array to achieve higher sensing performance. For example, Lv et al. [35] spin-coated a mixture of multi-walled carbon nanotubes and PDMS onto a stretched silicon substrate and demolded it after curing to obtain a wrinkled multi-walled carbon nanotubes/PDMS (MWCNTs/PDMS) sensitive layer structure. Two indium tin oxide/PET (ITO/PET) flexible electrodes are then sandwiched to form a typical “sandwich” structure, as shown in Figure 6D. The surface and cross-sectional SEM images of the sensor dielectric layer are shown in Figure 6E,F. The irregularly wrinkled structure of the sensitive layer increases the contact area with the electrode layer, so that the sensor reflects its extraordinary sensitivity and wide linear detection range. The response of the sensor to the pressure of 23 mg of rice and 2 mg of ultra-light millet is also shown experimentally, and the detection limit of the sensor can be as low as 0.2 Pa compared with the detection limit of 0.5 Pa of the commercial force meter. These excellent performances indicate the high resolution and accuracy of our sensor. In addition, Wang et al. [147] proposed a “sandwich” structure based on carbon nanotube paper film (CNTF) and stress-induced square frustum structure (SSFS) with magnetron-sputtered interdigital electrodes on PI, as shown in Figure 6G. The combination of CNTF fibers with a large specific surface area, a microstructured substrate with a large contact area, and multiple conductive paths of the interdigital electrodes results in a sensor with a large surface area. The sensor has an ultra-wide pressure detection range of 0.0003–200 kPa, with a sensitivity of 2027.5 k${Pa}^{-1}$ in the 0–20 kPa range and 174.7 k${Pa}^{-1}$ in the remaining range (see Figure 6H for details). Both of the above sensors can also be integrated into sensing arrays for applications in the field of tactile sensing.

#### 3.2.3. Single Electrode Structure

The single electrode structure has the advantage of having only one electrode grounded, greatly expanding the application of triboelectric sensing arrays in areas such as wearable pressure sensing devices, human-machine interfaces, tactile sensing devices, and intelligent identification systems. Single-electrode structures have been invented in which a single electrode acts as a triboelectric electrode layer and another triboelectric electrode layer acts as a potential reference that can be positioned anywhere in space. For example, in contact with the skin as a triboelectrical positive surface, it can detect micro-motions or collect energy. They are easy to construct and operate, have a simple design, are low-cost, and are easy to integrate with other structures. Most thin-film-based single-electrode triboelectric nanogenerators have a flexible triboelectric layer and electrodes for different sensing applications in addition to energy harvesting. Thus, these devices help to achieve wearability. For example, Lee et al. [149] developed an ultrathin lattice-based structured stretchable 8 × 8 array of single electrode structured triboelectric electrical sensing arrays, as shown in Figure 6J. The improved stretchability allows the device to be integrated directly into unevenly curved skin. The touch points are recognized, and the touch motion is tracked without an external power supply. Due to its favorable mechanical stretchability, the device can operate stably even when attached to human skin, showing its potential for wearable applications. In addition to this, Kim et al. [148] prepared a stretchable and biocompatible single-electrode triboelectric electrical sensing array based on plasticized PVC gel and graphene electrodes with the layer-level structure shown in Figure 6I. PVC gel is a suitable stretchable TENG dielectric material, and graphene is a highly conductive electrode. Thus, the graphene and PVC gel-based stretchable and biocompatible sensing array has excellent electrical output. Combined with its biocompatibility, stretchability, and stability, this self-powered skin-like triboelectric sensing array shows great potential for future wearable and biomedical applications.

### 3.3. Overview of Sensitive Layer Structures

Among the existing extraordinary-sensitivity flexible pressure sensing arrays [164], microgeometries are the most common method to obtain extraordinary sensitivity. However, they exhibit high nonlinearity when the applied pressure is high. The high degree of nonlinearity affects the determination of the degree of object deformation and increases the uncertainty between the output and the input. When pressure is applied to a sensitive layer containing inhomogeneous microstructures, initially larger microstructures are compressed, and as the pressure increases, smaller microstructures begin to be compressed. Thus, microstructures with a non-uniform distribution have higher sensitivity and better linearity than sensitive layers with periodic microarray structures under high pressure. However, various methods to improve linearity did not work well, and the definition of the high-pressure detection range differed from study to study. Therefore, the performance of sensors developed in different studies cannot be accurately compared due to the lack of standardized parameters [165]. The main advantages of porous structures in capacitive pressure sensing arrays are the large capacitance of the porous sensitive layer, the separation of conductive substances equivalent to a large number of electrodes, and the shortened electrode plate distance. As a result, the capacitance and sensitivity of the sensor are extraordinary. In piezoresistive and triboelectric sensing arrays, porous structures increase the contact area between plus two surfaces, thus effectively changing the conductive path [166] or enhancing the triboelectric effect [167], respectively, to improve the performance of sensing arrays. Meanwhile, to further improve the sensing performance of pressure sensing arrays, multi-scale hierarchical structures were developed, mainly including intrinsic hierarchical structures [168], multilayer-stacked hierarchical structures [156,169], and the combined hierarchical structure made up of multiple single structures [170,171,172]. In addition to this, the combination of microgeometric structure, porous structure, and multi-layer structure in the processing of sensitive layer structure with each other to form hybrid multi-structure is the current mainstream for improving the sensing performance of sensing arrays. We will discuss each of these four major structures in detail below.

#### 3.3.1. Microgeometric Structure

Microgeometric structure can be classified as periodic microarray structure [140,153,164], bionic structure such as: petals [163], leaves [173], fish scales [174], octopus tentacles [175], and irregularly wrinkled structure such as: sandpaper [161], spines [119]. These microstructures can also be deposited directly on flexible substrates (microstructured PDMS, PU, Ecoflex, silicon flakes, etc.). Meanwhile, a paper substrate with a natural microstructure, eco-friendly properties, and simple processing technology has been widely discussed at present. Therefore, paper-based flexible pressure sensors and sensing arrays have been widely reported [176,177].

Thus far, thirteen years ago, Bao’s group first proposed a flexible pressure sensor with a sensitive layer of micro-pyramid structure [178]. Compared with the flexible pressure sensor without the special structure of the sensitive layer, the sensing performance of this sensor, such as sensitivity and response time, was optimized in a linear manner. Subsequently, the micro-pyramid structure became the most widely studied and applied microstructure, and many methods for optimizing the micro-pyramid structure were proposed and gradually applied to the processing and fabrication of sensing arrays (for example, see [179]). Thouti et al. [151] prepared capacitive pressure sensors using micro-pyramid PDMS dielectric films. The effects of surface coverage and arrangement of the structure on the sensitivity of the pressure sensor devices were investigated, and the results showed that, in addition to the separation between micro-pyramids, their arrangement also has an effect on the sensitivity of the sensor. Meanwhile, the winkled structure can enhance the sensitivity of the sensor, and the micro-pyramid structure can also enhance the sensitivity of the sensor. Thus, Bae et al. [152] combined the two structures and developed a pressure/temperature sensor with a sensitive layer of surface-wrinkled pyramid microstructure (as shown in Figure 7A) for simultaneous real-time detection and differentiation of pressure and temperature stimuli. It also has a wide pressure detection range up to 25 kPa, a high linear pressure sensitivity of 0.7 kPa^−1^, and excellent repeatability in measuring temperature. It is also integrated into a 4 × 4 flexible sensing array for temperature and pressure differentiation experiments, and the results demonstrate the great potential of the sensing arrays in the field of flexible wearables. Moreover, Yang et al. [138] proposed an ultra-sensitive capacitive pressure sensor based on a porous pyramid sensitive layer structure (Figure 7B), whose porous pyramid structure SEM is shown in Figure 7C. Since the porous material has a lower compression modulus compared to the volume of the same material, the sensitivity is enhanced to an unprecedented 44.5 kPa^−1^ in the pressure detection range of 100 Pa. In addition, the contact area at the top of the pyramid is the smallest, resulting in a local concentration of stress at the top of the pyramid, which further reduces the effective compression modulus and is one of the reasons for the extremely extraordinary sensitivity of the sensor. In addition to this, the sensing array integrated into the resulting sensor exhibits insensitivity to strain forces and temperature, which is sufficient to prove its suitability for wearable device applications.

For periodic microarray structures, in addition to the aforementioned micro-pyramid structure, microdome and micropillar structures have also been developed to enhance the sensing performance of flexible pressure sensing arrays. For example, Lee et al. [156] proposed a flexible pressure sensing array based on a multilayer interlocking microdome-sensitive layer structure that can be designed with high pressure sensitivity and a linear response over an extremely wide pressure detection range. The multi-layered interlocking micro-dome structure increases the contact area inside the sensing array and is the main reason for the extraordinary sensitivity embodied in the sensing array. The interlocking stress distribution between the layers enables the sensing array to achieve a fast response over an extremely wide detection range. The single sensing unit is used to monitor respiration and pulse, and the integrated sensing array can monitor plantar pressure, indicating a promising future for wearable biomedical devices. However, in some harsh environments, the integrity of the sensing array can be compromised, and the layer structure can be displaced or even separated, which can significantly degrade the sensing performance of the array. To cope with such situations, Luo et al. [155] developed a flexible pressure sensor consisting of a sensitive layer of an inclined micropillar array structure (Figure 7F), whose SEM of the micropillar structure is shown in Figure 7D,E. The most outstanding property of this sensor is that it is not affected by damage such as scratches or cuts, and even the sensing segment cut off from the sensor will largely maintain the original sensing characteristics of the sensor. When the sensing array is stressed, the inclined micropillar structure bends rather than compresses, and the electrode spacing is more easily changed, resulting in a sensor with extraordinary sensitivity (0.42 kPa) and a very small detection limit (1 Pa). With such a powerful sensor, it is possible to cut the whole sensing unit into several sensing units and later integrate them into an array for some applications, such as tactile pressure sensing or normal operation under harsh conditions.

#### 3.3.2. Porous Structure

The sensing performance of flexible pressure sensing arrays based on planar-structure composite elastomers is poor. Materials with porous structures have the advantages of small density, large specific surface area, light weight, and strong elastic properties and are gradually being widely used in the fabrication of sensitive layers for flexible pressure sensing arrays. Foam structure, sponge structure, textile structure, and aerogel structure are the four most common porous structures for sensitive layers at present. Sensitivity is the most critical index to evaluate the sensing performance of a sensing array. To overcome the limitation of sensitivity, flexible pressure sensing arrays have been fabricated using high-cost materials and complex processing processes. For example, Li et al. [157] mixed multi-walled carbon nanotubes, zinc oxide nanoparticles (ZnONPs), and PDMS phases and used a sacrificial casting-etching method to obtain a nanoporous foam structure with ZnONPs as sacrificial materials to obtain flexible pressure sensing arrays that responded to a wide detection range and exhibited higher sensitivity when pressure was applied. In existing work on the fabrication of flexible pressure sensing arrays, porous sensitive layers are usually prepared by impregnating, depositing, and etching porous foams and sponges with micro- and nanoscale functional materials. For example, Dai et al. [182], inspired by the structure of bamboo, prepared flexible pressure sensors with extraordinary sensitivity and stability over an ultra-wide stress detection range by combining a unique composite material with a layered pore structure and a foam structure as the sensitive layer of the sensing array. It was also integrated into a five-layer sensing array to further validate the excellent stability brought by the sensitive layer structures, indicating its great potential in the field of electronic skin.

However, the process of preparing the above two porous structures is more complicated and costly. Therefore, they are not suitable for mass production of sensing arrays. In general, the flexible pressure sensing array we prepared should not only have fantastic sensing performance but also consider its manufacturing cost, process implementation, and many other factors. Thus, Chen et al. [158] prepared a MoS_2_/HEC/PU sponge flexible pressure sensing array by using a simple soaking method to coat a PU sponge skeleton with MoS_2_ and HEC to make a sensitive layer of the sponge structure. The resulting sensing array has excellent sensing performance, such as outstanding sensitivity, a wide pressure detection range, and a fast response or relaxation time. In addition, the integrated pressure detection array enables detection of pressure distribution, finger touch marks, and recognition of Arabic numeral writing. Due to its excellent performance and simple manufacturing process, it is expected to have great potential for electronic skin (the electronic skin is a soft, flexible sensor with integrated circuits that can also transmit electrical signals to the brain to allow the wearer to “feel” pressure, strain, or changes in temperature), health detection, and intelligent robotics.

In addition to foam and sponge structures, textile structures are now widely developed for applications in the fabrication of triboelectric sensing arrays. For example, Peng et al. [159] developed a flexible and stretchable all-nanofiber triboelectric sensing array electronic skin (e-skin). The top PLGA triboelectric layer, the bottom PVA substrate layer, and the middle AgNWs electrode layer all adopt an all-nanofiber cross-network structure and combine to form a three-dimensional micro- and nanoporous layered structure. As shown in Figure 7G, this structure provides a high specific surface area for the pressure-initiated electrical response and ensures the biocompatibility of the sensing array. The sensing array not only has extraordinary sensitivity and a wide detection range but also functions as an electronic skin with breathability and biodegradability. However, despite the extraordinary sensitivity and stability of this triboelectric sensing e-skin, there is and can be only a single monitoring of pressure or strain force. The goal of making sensing arrays is to improve their excellent sensing performance. In addition, we need sensors that can achieve simultaneous monitoring of multiple physiological indicators with limited capabilities. Thus, Wang et al. [180] developed a dual-mode electronic skin (Figure 7H) that integrates a pressure sensing layer and a temperature sensing layer in the same textile, allowing for accurate detection of pressure and temperature simultaneously. For pressure sensing, electrostatically spun PVDF/ZnONPs are used as the sensitive layers of the composite structure. For temperature sensing, carbon nanofibers (CNFs) with fantastic flexibility and electrical conductivity are used as the thermal resistance layer. In addition, conductive fabrics and PU films are selected as electrodes and insulating materials, respectively, to construct the whole flexible device. The electronic skin sensing array, which shows powerful capabilities in visual tracking and distinguishing pressure contours, shows its promising applications in intelligent robotics and health monitoring.

In addition, aerogels are commonly used as porous structures [160,183,184,185]. Liang’s team reported a multiscale aerogel with nanochannels within the cell wall (Figure 7I) [160]. Soft bottle-brushed polysiloxane was inserted into the MXene interlayer by covalent cross-linking, and the porous aerogel was formed by freeze-drying; its structural SEM image is shown in Figure 7J. Under pressure, the porous structure of the active substance and the layer spacing in the cell wall change simultaneously, resulting in a sensor with extraordinary sensitivity, a fast response time, and robust durability. The integrated 5 × 5 sensing array can even detect the pressure of mosquitoes during takeoff and landing, and such fantastic sensitivity indicates its great potential for sensitive tactile sensing and electronic skin applications.

#### 3.3.3. Multi-Layer Structure

In addition to the aforementioned microgeometric and porous structures, multi-layer structures are now widely developed to enhance the performance of sensing arrays. For example, Yang et al. [181] demonstrated a nature-inspired HB structure. Two microstructured hemispherical sensitive structure layers are superimposed on the electrodes, with the upper layer being a larger hemispherical microstructured PDMS layer and the lower layer being a smaller hemispherical microstructured rGO coated PDMS layer, and finally combined with the PEN substrate layer and PI substrate layer to form a sensing array with excellent sensing performance, as shown in Figure 7K. Compared with the single-layer structure sensing array, its sensitivity is improved by 14 times. The design of the multi-layer structure can improve the sensitivity of the sensing arrays; however, the signal crosstalk problem may occur in each sensing unit of the sensing arrays, which affects other excellent sensing performance such as the original sensitivity of the sensing unit. Therefore, Lin et al. [141] designed a multi-layer structure to overcome the crosstalk problem of sensing arrays. A PDMS film is used as an insulating layer to avoid the crosstalk of the two PVDF sensitive layers, which solves the crosstalk problem of the sensing array while substantially enhancing the sensing performance of the arrays.

#### 3.3.4. Hybrid Multistructure

We have already mentioned many layers of structure in the periodic microgeometric structure and porous structure introduced above. The finite element analysis experiments show that the layered structure controls the deformation behavior and pressure distribution at the contact surface, so that the contact area between the contact surface and the load increases rapidly and steadily, improving the sensing performance of the pressure sensing array. The microgeometric structure and porous structure can also substantially improve the sensing performance of the array. Therefore, layered structures generally do not appear alone in the sensitive layer structure of sensing arrays but in combination with microgeometric structures or porous structures in the processing and fabrication of multi-structured fused sensing arrays. For example, layered porous structures [160,171,172,182], layered periodic microarray structures [156], micropapillary structures [186], layered textile structures [169,187], and fine structures mimicking insect setae [168]. The interplay of porous and microgeometric structures is as mentioned above for micro-porous pyramid structures [122]. In addition to this, the sensitive layer is processed into some special structures to enhance the sensor’s sensing performance. For example, porous, wrinkled MXene sphere-based structures [188] and wrinkled structures [35,119].

### 3.4. Packaging Method

For flexible pressure sensing arrays, sensing performance is the primary factor we consider. We can improve the sensing performance of the sensing array by various methods, such as selecting different materials and processing different microstructures. However, the encapsulation of the sensing array is actually a crucial aspect for the sensing array to achieve excellent functionality, as the encapsulation not only protects the device at the physical level but more critically prevents the erosive effects of water, oxygen, and corrosive liquids in the external environment, etc. Nguyen et al. [189] proposed that the introduction of an encapsulation layer not only prevents physical damage and external environmental effects but also enhances the sensing array’s robustness and stability. There are two main approaches to packaging flexible sensing arrays: surface-applied packaging and bottom packaging. Surface-applied encapsulation is the process of attaching a flexible sensor to a protective film and covering it with another film to protect the sensing array from the external environment. This encapsulation method is suitable for light applications such as glove-based sensing arrays, sensing arrays on clothing, etc. However, this method is not suitable for some applications that require durability because it is less durable and easily damaged. Bottom encapsulation is the encapsulation of a flexible sensing array in a protective housing that protects the array from mechanical impacts, chemical corrosion, and other adverse environments. The advantage of the bottom package is that it is more durable and suitable for sensing arrays that need to be used in harsh environments, such as robotics, motion tracking, and other fields. Furthermore, this encapsulation method provides better mechanical support for the sensor, making it more stable and accurate.

Currently, the common packaging processes for flexible sensors include thermocompression, dispensing and coating, and silicone sleeve packaging. Thermocompression encapsulation of sensing arrays is a common encapsulation technique. Pressure is applied to the aligned device, and the encapsulating adhesive film is pressed onto the electrode layer at a high temperature. The high temperature and pressure cause the device to be completely encapsulated by the encapsulation film, creating a reliable water and oxygen barrier and maintaining extraordinary flexibility. Thermocompression packaging has the advantages of wide applicability and fantastic reliability. However, material compatibility and temperature and pressure control need to be considered. However, the high temperatures and mechanical stresses in thermocompression packaging can have an impact on the sensitivity, linearity, and stability of the sensing array. Dispense encapsulation of sensors is likewise a common type of packaging. Using a suitable adhesive, the sensing array electrode layer is dispensed and coated, and after cooling, an encapsulation layer is formed to protect the device from the external environment. The key issue in the dispensing method is adhesion, thermal conductivity, and other properties that should be considered when selecting an adhesive. The dielectric properties of the dispensing agent may affect the electrical properties of the sensing array, thus affecting the sensitivity of the sensor. The silicone sleeve encapsulation method for sensing arrays is similar to the dispensing package. It is intended to provide mechanical protection, dust and water resistance, vibration resistance, and isolation from electromagnetic interference for the device. However, the presence of silicone sleeves may lead to an increase in thermal resistance between the sensor and the environment, resulting in reduced sensitivity, a slower response, and other effects. In addition, some other encapsulation processes include cold welding encapsulation, spray encapsulation, laser encapsulation, and nano-printing encapsulation. In conclusion, flexible sensor encapsulation is an important technology to protect flexible sensors and extend their application fields. Choosing the appropriate encapsulation material, encapsulation form, and encapsulation method can further improve the reliability and stability of the sensing arrays.

### 3.5. Crosstalk Problem of Pressure Sensing Array

Pressure sensing array crosstalk problems are usually caused by physical or electrical coupling between adjacent pressure sensing units, resulting in measurement errors where the pressure measurements of one sensor are influenced by other sensors. Crosstalk problems may affect the accuracy and reliability of the sensing array. If sensors are wired too close to each other, electromagnetic interference may propagate through electrical coupling to adjacent sensors, resulting in electrical coupling crosstalk. In addition, vibration or pressure changes in one sensor may propagate to other sensors through physical coupling, causing mechanical coupling crosstalk. There is also the possibility that there is a large temperature difference between sensors and that thermal conduction may cause temperature changes from one sensor to propagate to other sensors, thus affecting their pressure measurements. To solve the crosstalk problem of pressure sensing arrays, anti-interference factors are considered in the sensing array design and layout. Increase the distance between sensing units or rationalize the wiring to reduce the effects of electrical and mechanical coupling. Appropriate isolation materials can also be used to solve the crosstalk problem. For example, Lin et al. [141] used a PDMS insulating layer to isolate the two electrode layers and eliminate the crosstalk problem in the array. Zhang et al. [190] used a similar approach to provide spacing for the pressure sensing array and achieved high spatial resolution of the sensing array for pressure imaging and ultra-low crosstalk. In addition, temperature compensation techniques were used to calibrate and compensate the sensing array for temperature to eliminate the crosstalk problem caused by temperature. In the future, researchers should not only work on methods to overcome crosstalk but also consider combining sensing arrays with advanced technologies such as machine learning and intelligent algorithms to process signals and achieve pressure sensing arrays at low cost, easy processing, low crosstalk, and fantastic reliability through rational design.

## 4. Applications

The above sections of this paper introduce the layer materials, structural design, and fabrication process of a flexible pressure sensing array. This part focuses on four aspects: physiological information, pressure sensing, haptic sensing, and other applications, and highlights the application profile of flexible pressure sensing arrays in skin sheets, showing their potential applications in electronic skin and wearable electronic devices.

### 4.1. Physiological Information Monitoring

Cardiovascular diseases kill about 1/3 of the world’s population each year [191], most of which can be prevented by long-term monitoring of physiological signals. With the rapid development of flexible sensing technology, today’s wearable pressure sensors are capable of non-invasive and continuous acquisition of physiological signals in a reliable manner [192,193,194,195,196,197,198], which is a major advancement for the field of healthcare.

Pulse is an important part of physiological information, and the measurement of pulse information can reflect the health status of the body, such as arrhythmia, exercise status, and sleep status. The doctor has to judge the pulse signal generated by the radial artery in the arm during the diagnosis, but the diagnosis results that depend on the experience of medicine are uncertain [199,200,201,202]. We need to consider practical applications when designing sensing arrays for measuring pulse signals. One is that the radial artery is located deep in the skin [203], and the signal is weak and not easy to measure, so the sensor must have fantastic sensitivity and interference resistance. In addition, because the radial artery is small (the typical outer diameter is 3–4 mm), conventional sensors usually require the operator to find the best pulsation point to place in order to measure accurately, which relies on the operator’s experience. The proposal of spatiotemporal pulsation provides a new way of thinking to solve the above problems.

Bai et al. [204] proposed a wearable device that enables spatiotemporal measurements of arterial pulse waves and consists of 10 × 10 thin film transistor (TFT) arrays and piezoresistive sensor sheets with interlocked microstructures. They cleverly used a transistor material with active matrices to overcome the inevitable crosstalk problem of passive matrix sensing arrays, allowing the device to achieve two-dimensional (2D) signal mapping at high frequency with minimal crosstalk. Moreover, they modulated the operation voltage of the TFT so that the device can simultaneously achieve a fantastic sensitivity of 16.8 kP$a^{-1}$ and a low power consumption of $10^{1}$ nW, which provides excellent conditions for spatiotemporal pulse wave measurements. As shown in Figure 8A, when the device is attached to the wrist, the pressure generated by the radial artery can cause the sensing array to output a spatiotemporal pulse wave map, where the higher voltage output curve is the actual waveform closest to the arterial line, and the location of the pulsation and the accurate pulse image can be obtained from the map. This device well overcomes the limitations of the traditional pulse sensor with single-point measurement and has great potential for future applications as an intelligent medical device.

Zheng et al. [205] proposed a low-crosstalk stretchable ionization sensing array for wrist pulse measurement, as shown in Figure 8B, which achieves excellent performance indexes such as a fantastic sensitivity of 521.69 kP$a^{-1}$ and a broad linear sensing detection range of 200 kPa using the electric double layer (EDL) effect of MXene (${Ti}_{3}C_{2}T_{x}$)/robust aramid nanofiber (ANF) electrodes, ionic film, and fiber spacer layer. Moreover, they also used the serpentine island bridge structure to increase the stretching of this array and effectively reduce the crosstalk problem of the array. They integrated the device with back-end circuitry and attached it to the volunteer’s wrist to obtain pulse waveforms at multiple locations, from which the exact pulse signal and the location of the artery can be accurately obtained. The device is expected to be a medical diagnostic device for the treatment of cardiovascular diseases.

The pulse measurement device can also be combined with the Internet of Things for telemedicine diagnosis, which only requires a physician on the computer side to diagnose the patient’s physical condition and help patients achieve home treatment. Recently, Liu et al. [206] proposed an intelligent diagnostic system for monitoring temporal pulse, as shown in Figure 8C, which allows the doctor to diagnose the patient through the pulse signal transmitted wirelessly from a remote location. The intelligent system includes a multichannel signal processing circuit, intelligent algorithms and software, and a multichannel flexible pulse pressure sensing array based on a sandwich structure, in which the sensing array consists of two PI and Au electrode layers, PDMS support layers, and multiwalled carbon nanotube polydimethylsiloxane nanocomposite (MPN) cells. Thanks to its pyramid-like bilayer microstructures with different upper and lower surfaces, the array achieves a wide detection range of 0–18 kPa and a fantastic sensitivity up to 35.02 kP$a^{-1}$. They have successfully integrated three 3 × 3 sensing units into the narrow position above the radial artery, which can collect pulse signals from three positions of Gun, Guan, and Chi, and the pulse signals can be transmitted to the cloud terminal after processing by intelligent algorithms so that the medicine can carry out diagnosis and treatment. This intelligent diagnosis system can realize three links: pulse measurement, pulse analysis, and disease diagnosis at the same time, which fills the gap in pulse analysis and diagnosis in the previous work field. The device is of great significance to the future application and development of remote diagnosis and wearable health devices.

In addition to pulse signals, the measurement of respiratory rate is also important for human health monitoring, which has a great role in cardiovascular diseases, lung diseases, apnea syndrome [208], and daily physical exercises. Fan et al. [207] reported a washable friction electrode sensing array fabric, as shown in Figure 8D, with two TATSAs integrated into a shirt for the monitoring of pulse and respiratory signals in real time, the acquisition of which not only helps to assess the respiratory status but also helps to diagnose coronary heart disease and apnea syndrome by comparing it with the respiratory heartbeat signal of a healthy person. A high or low respiratory rate is a warning of possible abnormalities in the body; the development of this device provides an efficient and convenient way to monitor physiological information on a daily basis. It is important to note that the sensing array should also have superior flexibility and anti-interference capability so that the measurement information is not affected by external interference during daily wear.

Blood pressure is also one of the most important parameters of physiological information, and deaths due to hypertension account for 5.12% of global deaths. Early detection and prevention of hypertension can effectively reduce the detection and mortality of cardiovascular and other related diseases [209], which is of great importance to our health. Traditional blood pressure measurement is usually performed with a cuff tied to the arm using a pump sphygmomanometer [210], which is not portable and does not allow for long-term monitoring of blood pressure. The advent of flexible electronics has solved these problems.

Vega et al. [211] proposed a novel epidermal electronic device, also called multiPANI, that enables cuffleless blood pressure (BP) monitoring. The array consists mainly of a PET film layer and a PANI layer and is encapsulated with biocompatible silicone elastomer (Ecoflex^TM^, Smooth-On) and a medical patch (Tegaderm, 3M). They combined the array with wireless-flexible hybrid electronics to create an integrated wearable device, which was placed on the radial and ulnar arteries of the wrist to capture the precise pulse wave velocity (PWV), and then converted the PWV to a PWV by building a BP Estimation Model. The device has successfully achieved long-term continuous blood pressure monitoring, but the sensing array needs to be positioned in a customized way for different people in order to accurately capture the pulse signal, which has some limitations.

The novel bioimpedance (Bio-Z) sensing array that can be used for uncuffed blood pressure monitoring by Ibrahim et al. [212] addresses this problem. They use an unsupervised machine learning method that combines pulse signals from multiple sensing locations collected to encode and reconstruct them, and this learning method can continuously adjust the data processing by continuously learning the function between the artery and the sensor through training data to achieve BP prediction independent of location sensing. It is easy to see that these sleeveless blood pressure measurement methods rely not only on the acquisition of accurate pulse signals but also on the construction of the BP Estimation Mode. In future work, the advantages of both should be combined to develop intelligent electronic products that can be put into practical applications.

Physiological information monitoring has been a popular research area in wearable sensors, where sensors are placed on the skin surface of the human body to record pressure changes and thus respond to physiological information. In future work, this gap in collected signal analysis as well as disease diagnosis should be bridged so that the sensing arrays can provide telemedicine-assisted treatment while making full use of their excellent performance.

### 4.2. Motion Recognition

Timely detection of the status of human joints [213] or non-joints [214] can effectively prevent sports injuries and also assist in the rehabilitation of patients, and real-time monitoring of various body parts to obtain information is of great significance for human health monitoring. There are many parts that each person needs to move frequently in daily life, such as the knee joint, elbow joint, eye muscles, head, throat, etc. The movement status of each part can reflect the health status of different parts of the body.

This section will be divided into two parts to discuss the motion detection applications of joint parts and non-joint parts, respectively, and demonstrate their potential applications in the field of motion recognition.

#### 4.2.1. Joint Areas

With the improvement of social and economic levels, people gradually start to pay attention to the health condition of their bodies. As a sport with low hardware equipment requirements and small costs, running has naturally become one of the most popular sports, but long-term improper running posture may cause knee injury or even meniscus wear [215], and this injury usually needs to be detected when the relevant examination is conducted in the hospital, so preventive measures for knee injuries are necessary. Wang et al. [216] proposed an ionotropic sensor with fantastic flexibility that was prepared as a sensing array that could cover the whole knee and accurately map the spatial pressure distribution of the knee. As shown in Figure 9A, the sensor consists of ionic liquid/thermoplastic polyurethane (IL/TPU) nanofiber electrolyte and two fiber nanoelectrodes, and TPU fabricated by electrostatic spinning technology endows the sensor with a porous structure and a mesh structure, which lays the foundation for the realization of the superior performance of the sensor, and the device achieves not only wearing comfort but also fantastic sensitivity, a wide detection range, and other performance indicators. They made the sensor into a 5 × 5 sensing array fabric, which can be worn on the knee to accurately respond to the pressure distribution map of the knee in different postures during running, an application of great significance for the prevention of running knee injuries. If the data collected is combined with network technology, it can also send an alert when the knee is injured, which can be used not only for daily health monitoring but also to assist patients in limb recovery therapy.

In the same year, Wang et al. [217] also proposed a wearable haptic sensor, also based on a framework of all-nanofiber networks, which can be customized to any size and adapted to various parts of the human tissue. As shown in Figure 9B, a sensing array was prepared by distributing eight sensors at 45° intervals on an elbow pad, which was worn on a volunteer to collect pressure signals at different angles of elbow flexion and to monitor the state of the elbow joint in real time to prevent muscle strains during exercise. In addition, they also made a 6 × 6 sensing array, which can be attached to the arm to map the pressure distribution corresponding to different states of the hand and perform simple gesture recognition. This device has great potential for future applications in the fields of motor rehabilitation and human-machine interaction.

Zhong et al. [218] reported a textile-based flexible piezoresistive sensing array that can be worn on ankles, knees, shoulders, and other joint sites to map their pressure. The flexible sensing array was woven from HCPYs, and during its fabrication, they cleverly optimized the microporous structure by varying the humidity and exposure to improve the performance of the sensor. Moreover, the material maintained excellent comfort, breathability, and 3D adaptability both alone and compiled as a fabric, and these superior properties allowed it to be used as a wearable device. As shown in Figure 9C, the HCPYs are sewn into the elbow joint of the volunteer sportswear to detect the spatial pressure distribution in that area. All of these sensing arrays for measuring joint areas mentioned above are expected to become wearable devices in daily life in the future.

In addition to the elbow and knee joints, the neck is also important for human health, and long-term poor posture (e.g., ambulation, poor sleeping posture) can induce cervical spondylosis. People’s excessive use of electronic products has led to an increase in the prevalence of cervical spondylosis year by year. Zhang et al. [219] proposed a stretchable capacitive sensing array; as shown in Figure 9D, the array mainly consists of liquid metal film electrodes with a three-dimensional icicle-shaped structure. The top layer of an icicle-shaped film electrode array is placed orthogonally with the bottom layer of flat film electrodes to form the sensing array, which can maintain fantastic sensitivity under high pressure due to its special icicle structure and can still sense the pressure accurately under fantastic stretching conditions. Wearing the 4 × 4 sensing array on the neck can reflect the pressure distribution in different postures in this area. If this device is combined with deep learning, it can be used to remind people to change unhealthy cervical posture, and the device is also expected to be used as an adjunct to rehabilitation training for patients with cervical spondylosis.

Motion detection of joint parts also has great potential in the field of gesture recognition. As early as 2005, Lorussi et al. [222] proposed a smart glove that could detect finger posture and motion. Chouhan et al. [223] have integrated a computer and a wired interactive glove into a smart system that can accurately recognize and map hand gestures, transmit the data to a computer, and combine machine learning to translate sign language into a language people can understand. In addition to smart gloves, Ferrone et al. [224] developed a wearable bracelet that can be used for gesture recognition and consists of cloth and strain sensing arrays, where the strain sensors are not in direct contact with human skin, and machine learning is also used to process the signals collected by the sensing array. In the future development of wearable devices related to smart gloves, in addition to the user’s long-term wearing comfort, breathability, and accurate characterization of sensor recognition, they should be combined with advanced technologies such as machine learning to develop intelligent systems with diversified functions, such as assisting patients in hand movement rehabilitation, sign language recognition, and controlling computers using smart gloves.

#### 4.2.2. Non-Joint Areas

In addition to the detection of joint parts, the monitoring of the motion status of non-joint parts is also of great importance. For example, the fatigue state of a person can be determined by the frequency of blinking, and Xu et al. [129] reported an integrated sensor patch that can collect the pressure signal from the eye. The patch consists of a strain sensor and a pressure sensor, in which the strain sensor is constructed with a snake-like structure using a conductive composite sensitive material and the pressure sensor is made of a rigid piezoelectric ceramic PZT-5 material. This ingenious design makes the patch not affect the acquisition and performance of the sensor signal under stretching, twisting, and bending, which provides a fantastic basis for it to capture the weak signals from the skin surface. As shown in Figure 9E, they integrated a 3 × 3 piezoelectric array and strain sensor, which can be worn in the corner of the eye to accurately acquire the motion state of the eye muscles. If the voltage information collected by this sensing array can be correlated with the blink frequency during future research, this sensing array is expected to be a wearable device that can monitor visual fatigue.

Real-time acquisition of the motion state of the larynx during speech can be used in the field of speech recognition. Zhu et al. [220] proposed a self-powered, flexible electronic skin sensing array. As shown in Figure 9F, the array is integrated by a single-electrode triboelectric sensor and a piezoelectric sensor, which are made into a 3 × 3 cm sensing array that can detect different output signals corresponding to speech and silence when attached to the position of the throat. This has great potential for future applications in the fields of speech recognition and vocal rehabilitation exercises. The array can also reflect the different voltage output images corresponding to head nodding and head shaking, and these output response images may be able to assist in the diagnosis of Parkinson’s disease in the future. The application of sensing arrays to the larynx requires consideration of whether swallowing motions will have an effect on signal acquisition, so future research should take these unavoidable interferences into account.

About 1/3 of a person’s life is spent in sleep, and fantastic regular sleep habits help to relieve physical fatigue and are important for human growth and development and immunity enhancement [225]; conversely, poor long-term sleep may lead to poor mood [226], memory loss [227], and even cardiovascular disease or even life-threatening [228]. Real-time continuous monitoring of sleep status is a proven method for assessing human health, and many researchers have worked in the past to develop devices such as eye masks, belts, and local patches to obtain limb or eye movements and breathing status to assess sleep status [229,230,231]. However, these devices may produce a constricting sensation that affects the quality of sleep, ensuring the comfort and accuracy of the devices. In order to ensure the comfort and accuracy of the device, two perspectives can be considered to solve the problem: one is to develop a sensing array with comfort that can directly fit the skin, and the other is to integrate the sensing patch with common fabrics. Yue et al. [232] proposed a smart electronic skin with excellent breathability and comfort. The electronic skin consists of a triboelectric (pressure-sensing) layer, a temperature-sensing layer, an ant antenna structure, and the triboelectric collection system, which can simultaneously detect multiple pressure, temperature, and humidity levels. The e-skin achieves fantastic sensitivity while maintaining excellent biocompatibility and comfort, which has great potential for future wearable devices.

Kou et al. [221] developed a smart pillow that can monitor the motion state of the head during sleep. As shown in Figure 9G, it is composed of a 5 × 12 array of f triboelectric sensors and a regular pillow placed under it. The triboelectric sensing array is composed of Kapton film, porous PDMS, and aluminum (Al) electrodes, and the porous PDMS has fantastic breathability and comfort. The smart pillow can give early warning to the behavior of infants and elderly people falling from the bed during sleep, and the system will sound an alarm when the pressure stimulus falls on the five triboelectric sensors at the edge of the pillow.

Although all the above-mentioned sensing arrays have shown superior performance, there is still much room for improvement as portable wearable devices in daily life. In the above applications, some authors have integrated the sensing arrays into clothing, which can not only maintain the accuracy of the signals collected by the sensing arrays but also take into account the comfort of users. This design idea can be used in the production of future wearable devices. Secondly, considering the signal acquisition of sensing arrays used for motion detection, how to integrate these links into portable wearable devices should also be a problem that future researchers need to solve. In addition, wearable sensing arrays for motion detection should be combined with advanced electronic technologies to develop more functions, such as combining with cell phone APPs for timely alerts of injured body behaviors and continuous monitoring of body part information to assist in sports rehabilitation.

### 4.3. Haptic Perception

The development of haptic sensors is indispensable for today’s robotics, prosthetic fabrication, electronic skin, and human-machine interaction. Haptics is an important sensory function for humans to perceive their environment [233], and signals (e.g., size, shape, hardness) generated by various external stimuli (e.g., temperature, humidity, pressure) can be mapped to the brain through touch [1]. The fantastic sensitivity and resolution of simulated human skin are key elements in designing flexible sensing arrays for haptic perception.

Doctors can touch a patient’s body part to make an initial judgment of the patient’s condition by sensing the softness of that part and can also determine the extent of the surgical incision during surgery based on the different hardness of the body part [234,235,236,237,238], but doctors cannot determine the exact extent of a lesion by touch alone. The development of haptic-aware electronic devices has solved this problem. Cui et al. [239] proposed a large-area haptic glove with 112 force-softness bimodal sensors (FSSs), which can be worn on the human hand to identify the physical features of the patient. One of the FSSs is composed of a piezoresistive pressure sensor with a protrusion on the top, and the structure of the FSSs is shown in Figure 10A. This spherical protrusion structure allows the sensor to better map the softness of the sample. To ensure that the tactile glove can accurately sense pressure and have accurate softness mapping, the sensor electrodes are designed using a common grounded single channel wiring method that can be adjusted to collect information from each sensor on the glove and effectively overcome the crosstalk problem of the sensing array signals. The combination of this tactile glove and the machine learning method allows for the initial assessment and diagnosis of patients by touching and pressing four parts of the human body (arms, chest, abdomen, and shoulders) and has an accuracy of up to 0.98 in identifying the specific areas of neck and back pain in the four mannequins, which is an improvement of about 0.6 probability over the accuracy of manual diagnosis. 

In the preparation of sensing arrays, in addition to extraordinary sensitivity and resolution, mechanical crosstalk and electrical signal crosstalk are also unavoidable problems. Thus far, there is no method that can completely overcome mechanical crosstalk. In future work, not only the structure of the sensing array should be modified, but also the sensing array should be combined with the back-end processing equipment to compensate for this limitation through data analysis so that the sensing arrays can have higher accuracy in recognizing body features.

### 4.4. Human-Machine Interaction

Human-machine interaction (HMI) has also become a popular field in recent years. The integration of flexible pressure sensing arrays with wireless communication can produce systems that can realize human-machine interaction, which has a wide range of applications in speech recognition and gesture recognition [1,243]. For example, gesture recognition can be used to send some commands to the computer to make corresponding operations. In this subsection, we will discuss the skin-surface-based pressure sensing array from the field of human-machine interaction.

Zhang et al. [214] proposed a wearable pressure sensing array that uses a layer-by-layer assembly of mesh-like micro-convex PDMS, MXene/Ag nanoflowers (AgNFs), and a flexible interfinger electrode as a force sensing layer, a force transmission layer, and a force receiving layer, respectively. The mesh-like micro-convex structure of the array can effectively increase the size of the contact area, which plays an important role in improving the sensitivity of the sensor. The mesh-like micro-convex structure of the array can effectively increase the contact area, which plays an important role in improving the sensitivity of the sensor. They cleverly introduced AgNFs into MXene nanosheets, which act as a “bridge” connecting each layer of MXene nanosheets to enhance electron transfer effectively. Thanks to the excellent structure of the sensing array, the sensor achieves excellent sensitivity up to 191.3 kP$a^{-1}$ and a fast response and recovery time, which provide fantastic conditions for its preparation as a sensing array. As shown in Figure 10B, the authors integrated five flexible pressure sensors, a microcontroller, and a wireless Bluetooth module into a smart glove and put the glove on the user’s hand, which can detect the degree of bending at the joints of five fingers, and the various gestures made by the wearer can be reflected on the robot hand model built by the computer through human-machine interaction. This application can not only be applied to the future field of human-machine interaction but can also be used not only in the future human-machine interaction field but also in the fields of assisting patients in motor rehabilitation, designing intelligent prostheses, and robotics.

In addition, a haptic sensing array can be combined with electronics to realize a haptic sensing system that allows human-machine interaction. Jiang et al. [244] proposed a thin conformal wrinkled graphene-elastomer composite material that can overcome the problem of conventional graphene, which is prone to fracture under certain conditions and has limited stretchability. Jiang et al. proposed a thin conformal wrinkled graphene-elastomer composite material that can overcome the problem of conventional graphene’s tendency to fracture under certain conditions and limited tensile properties and used it to fabricate a triboelectric sensing array. The array exhibits excellent strain insensitivity due to the unique conformal pleated structure of the material and also has superior stretchability, maintaining a stable signal output even after stretching to 100% strain. These superior properties provide fantastic conditions for the fabrication of electronic skin, which can be attached to the human skin to deform with the uneven surface of the skin but not affect its output. The authors propose a system that enables human-machine interaction and consists of a sensing array composed of nine friction sensors, signal acquisition and back-end circuitry, and a computer. When a finger touches the array with two different forces, the device can recognize the pressure and position of the finger’s touch. At the same time, the sensing array can also send out different friction electronic signals according to the different positions of the finger touch and realize the control of the computer by the response and sequence of the different positions. This sensing array can be used in future work to diversify the application of human-machine interaction by combining it with intelligent electronic products, etc., and sending commands to them to perform operations through the array. Of course, the most difficult and important part of this process is how to prepare the sensing array into a wearable electronic skin that can be put into real life.

Human-machine interaction has brought many conveniences to users, and the rapid development of artificial intelligence in recent years has put more demands on human-machine interaction systems, which also need to consider issues such as functionality, safety, portability, and ease of operation [245]. For example, user habits can be considered to be incorporated into human-machine interaction systems to improve the safety of their operations. Advances in the field of artificial interaction will play a significant role in the future development of intelligent electronic products [246].

### 4.5. Other Epidermal Applications

In addition to the above-mentioned applications, there is great potential for other applications of skin-surface-based pressure sensing arrays. In this section, the training and rehabilitation ball, wearable fitness cap, and ionic liquid sensing array that can monitor the body pressure of patients in beds and wheelchairs are discussed. They showed the great potential of sensing arrays in the field of wearable devices and medical applications.

Regular rehabilitation training plays an important role in the treatment of Parkinson’s disease patients and can effectively improve muscle control and motor ability [247]. However, most of the traditional rehabilitation training centers rely on bulky and large equipment, which requires the collaboration of medical and nursing staff to perform training, which is expensive and has limited training scenarios. Yao et al. [240] proposed a training ball that can be used for motor rehabilitation of Parkinson’s patients, as shown in Figure 10C, which consists of a piezoresistive sensing array on the surface and an internal communication wireless chip. It can be used to monitor the recovery of patients in real time by quantifying the pressure as a rehabilitation effect through a WeChat app in a timely manner. The development of this device has important implications for the hand muscle training of Parkinson’s patients.

Loose or ill-fitting helmets on the heads of athletes may lead to traumatic brain injury (TBI) [248]. Helmet manufacturers believe that the fit of the helmet to the head is also an important factor in the safety performance of the helmet, but there is no device that can detect information related to people’s heads. To solve this problem, Masihi et al. [241] proposed a porous PDMS capacitive pressure sensor; they introduced nitric acid in PDMS and sodium bicarbonate species to prepare a PDMS porous dielectric layer using chemical reaction, and this method can provide more reliable and uniform pore distribution. They also optimize the dielectric constant and porosity of the PDMS layer by changing the concentration of each substance and curing temperature, which provides a fantastic condition for the sensor to achieve excellent performance. As shown in Figure 10D, a wearable fitness cap consisting of 16 pressure sensing arrays was also developed to reflect the head pressure of players when wearing helmets. This device can provide feedback on the correct information of the head to wear a more suitable helmet during athletes’ games and sports, which can effectively prevent head injuries from improper helmet fit and is of great significance to the development of the wearable device field. In addition, other protective equipment worn by athletes during training can also be developed with corresponding sensing arrays to correctly reflect the pressure information of each part of the body, thus preventing athletes from being injured due to equipment mismatch.

Pressure injuries are common in bedridden and wheelchair-dependent patients and are usually caused by wounds caused by prolonged pressure on the skin and bones in the same position, which is not conducive to rehabilitation [249,250]. Therefore, regular position changes are needed to prevent injuries [251], but position changes are not only related to the mattress and injury-prone areas [252]. Frequent changes may also cause further injury to the patient. To solve this problem, Han et al. [242] proposed a battery-free wireless ionic liquid sensing array, as shown in Figure 10E, which consists of a near-field communication (NFC) antenna, a thermistor, and a pressure sensor printed on a flexible circuit board. The sensing array, designed according to the different curvature and area of each body part, is installed at different locations (including the sacrum, elbow, and heel) to continuously monitor the pressure and temperature changes of the skin when the body is lying down in real time. The sensing array not only overcomes the limitations of the traditional pressure sensor single-point measurement but also adopts an open layout to increase breathability for long-term monitoring. In addition, the array can also be placed on the sacrum, sciatic bone, and heel to monitor pressure and temperature changes while the patient is moving in a wheelchair, and the development of this device has an important role in the diagnosis and prevention of pressure injuries.

From the above applications, it can be seen that the skin-surface-based pressure sensing array can not only integrate a well-functioning intelligent system in combination with back-end processing equipment but also be designed to monitor pressure and other skin-surface information (temperature) simultaneously, which can provide a more accurate and comprehensive response to the body’s condition. In future research efforts, multi-functional sensing arrays should be developed to achieve multi-dimensional information collection, not limited to pressure measurement, so that we can continue to progress in the field of wearable devices.

## 5. Integration with Back-End Circuits

Flexible electronic devices, which enable the monitoring of physiological signals on the surface of human skin, have become a revolutionary technology in the fields of sports health management, disease diagnosis and monitoring, environmental monitoring, and human-machine intelligence interaction. In fact, to meet the demand for health monitoring and medical diagnosis, many researchers have already combined flexible physiological sensing arrays (e.g., strain/pressure sensing arrays, electrophysiological signal sensing arrays) with back-end wearable health monitoring systems (and back-end circuits) to develop wearable device platforms. This is important for a comprehensive diagnosis of human health while monitoring body parameters and health-related conditions. A wearable health monitoring system, as an important part of the wearable electronic device platform based on a flexible pressure sensing array, is a key development direction and solution strategy.

The signal at the front end is processed by a back-end circuit, filtered, amplified, and digitized, and converted into a digital signal that can be processed directly by a microcomputer. In the following, we will elaborate on the existing technologies for filtering, amplification, and digitization. Filtering is the first-stage component of the back-end circuit. The sensing signal is influenced by the external environment and generates interference signals. Without the processing of the filtering circuit, the interference signals can seriously affect the final result. Classified by component composition, they can be divided into active and passive filters. Active filters have high gain, anti-attenuation, high input impedance, and low input impedance, which cause less interference to the input signal. Signals from sensing arrays are concentrated in the low frequency range, and low-pass filter circuits are often used. Commonly used active low-pass filter circuits are single-pole low-pass filters, Sallen-Key low-pass filters, and cascaded low-pass filters. Single-pole low-pass filters are simple and intuitive in their structure. The Sallen-Key low-pass filter is the most commonly used low-pass filter with a higher attenuation capability for medium- and high-frequency signals. However, the structure is complex, the power consumption is relatively high, and the implementation cost is high. The cascaded low-pass filter consists of multiple single-pole low-pass filters and Sallen-Key low-pass filters, and its ability to attenuate medium- and high-frequency signals increases with the number of cascades. Amplification is the second component of the back-end circuit and can be classified as bipolar junction transistors (BJTs) or field effect tubes (FETs) according to the transistor composition of the amplifier, which has a simple structure, mature technology, low cost, and is widely used. However, when the amplifier is not amplifying the signal, the bipolar junction transistor has static power consumption, which makes it difficult to integrate large-scale circuits and achieve low power consumption. FETs do not have static power consumption and can be integrated on a large scale, and the area occupied by the circuit is small. However, the circuit structure is complicated, and the cost is high. The circuit structure of an amplifier can be divided into common source amplifier circuits and differential amplifier circuits. The common-source amplifier circuit, which uses bipolar junction transistors, can realize the coupling of multi-stage circuits. Digitization is the third-level component of the back-end circuit; the sensing signal is analog, microcomputers cannot directly handle analog signals, only digital signals, and the digitization of the signal is an essential part of the sensor. There are many mature technologies for digitizing signals, such as successive approximation (analogue-to-digital conversion) ADCs, integral ADCs, parallel comparison (analogue/digital) A/D converters, voltage-to-frequency converter ADCs, and pipelined ADCs. Succeeding approximation ADCs have sampling rates up to 1 MSPS and low power consumption. However, at resolutions higher than 14 bits, the cost is higher. Integral ADCs, with resolutions up to 22 bits, have low power consumption and low costs. However, the conversion rate is low, ranging from 100 to 300 SPS at 12 bits. parallel comparison A/D converters, due to their parallel structure, have a high conversion speed. However, it has high operating power consumption and sacrifices conversion accuracy for speed. The voltage-to-frequency converter ADC, which is mainly used in medium- or lower-speed, medium-accuracy data acquisition and intelligent instruments, has high accuracy and low power consumption. However, its conversion rate is limited to 100–300 SPS at 12 bits. Streamline ADCs provide high-speed, high-resolution analog-to-digital conversion with very low power consumption and a small chip size. As for the connection between flexible and rigid back-end circuits, let us take the example of the formation of a semi-flexible printed circuit board. First, a layer of copper foil is placed on the flexible substrate, then a layer of corrosion-resistant or lithography-resistant material is printed on the desired part, and finally, the unwanted copper foil is removed by chemical etching or lithography, thus creating the desired circuit structure. After etching out the circuit structure, the chips and circuit components to be soldered are placed in their corresponding positions and soldered to the flexible circuit board with solder paste.

However, when it comes to integration with back-end circuits, many researchers’ development directions are mainly focused on building wearable integrated device platforms by monitoring chemical molecules in the human body. Compared to monitoring chemical molecules in the human body, integrated wearable devices that monitor human physiological indicators are underreported. The physiological signals of the human body include pulse beat, heartbeat, breathing, and other signals. The flexible pressure sensing arrays reviewed in this paper are commonly investigated for monitoring physiological signals. Therefore, the integration of flexible sensing arrays with conventional rigid circuits, with semi-flexible circuits, and with fully flexible circuits based on the aspect of physiological signals is discussed in detail below.

### 5.1. Integration with Conventional Rigid Circuits

The prepared flexible sensing array is combined with a printed circuit board (PCB) to form a complete flexible wearable device. The importance of flexible sensing device integration is reflected by pairing mobile devices via wireless communication technology (Bluetooth or NFC) for real-time data collection, transmission, and analysis, as well as the ability to remotely share results and receive health guidance from physicians in hospitals in real time. Some physiological signals of the human body are also often monitored to reflect the health of the body, and pulse rate and heart rate have received much attention as the most typical human physiological signals. For example, Kaisti et al. [253] developed a flexible wearable wristband based on an array of micro-electro-mechanical system (MEMS) elements integrated with a back-end system for continuous recording of arterial pulse for remote monitoring and personalized treatment of cardiovascular diseases. In addition, Polat et al. [254] proposed a graphene nanosheet-based epidermal motion sensor to estimate the swallowed volume of liquids (Figure 11B,C). The integration of an epidermal graphene-based piezoelectric sensor with conventional rigid electromyogram (EMG) electrodes into a sensor system improved the accuracy of estimation and was able to distinguish swallowing movements from motion artifacts. In addition, a novel machine learning algorithm is proposed, resulting in 92% high accuracy to discriminate swallowing of 5–30 mL of water. This system platform is expected to be widely used in sports medicine, rehabilitation, and the detection of neonatal swallowing dysfunction. However, in practical applications, the sensing arrays suffer from structural mismatch and integration bias. In particular, the arrays attached to the skin experience different bending strains and thus suffer from uneven pressure ringing. Therefore, to solve these problems, Kim et al. [255] proposed a new circuit approach to obtain uniform pressure response in nanomaterial-based skin flexible pressure sensing arrays. An important contribution was made to the application of pressure sensing arrays to wearable devices.

In addition, Liu et al. [206] proposed a 27-channel flexible pulse sensing array based on a “sandwich” structure pressure sensor (Figure 11A) and connected it to an integrated back-end circuit to realize a wearable system for remote disease diagnosis. Moreover, two sets of supporting intelligent algorithms were developed. They were used to extract six-dimensional pulse information for medical diagnosis and to achieve automatic pulse recognition, respectively, with a recognition rate of 97.8%. Thus, it shows that this multi-channel disease diagnosis system has great application prospects in the healthcare field. However, biocompatibility, the primary consideration for flexible sensing devices, has an insurmountable gap with the PCB. As a result, researchers have gradually started to favor the development of biocompatible flexible printed circuit boards (FPCBs) to replace conventional rigid PCBs.

By utilizing commercial or semi-commercial back-end electronics, efficient filtering, amplification, and digitization of the sensing array output signals can be achieved to meet the requirements of specific applications and to obtain extraordinary-quality signal data. First, back-end electronics are typically equipped with integrated analog-to-digital conversion modules for converting analog signals to digital form. By configuring its resolution and sampling rate, accurate digitization of the sensor signal can be ensured. If the sensing signal is weak, the signal can be amplified using an adjustable gain amplifier in the back-end electronics. Depending on the sensitivity of the sensing array, the gain value of the adjustable gain amplifier is adjusted to bring the signal to an optimal level. Afterwards, the filter module in the back-end electronics is used to filter out noise or unwanted frequency components from the sensor signal. Finally, in order to transfer and interact with the sensing array for data sharing and remote control, the back-end electronics usually provide various communication interfaces such as Universal Asynchronous Receiver/Transmitter (UART), Serial Peripheral Interface (SPI), Inter-Integrated Circuit (I2C), Ethernet, or wireless protocols. In addition, flexible wearable devices that integrate back-end circuitry with front-end sensing arrays are subject to various kinds of interference from the outside world in their applications. When crosstalk is a problem in the sensing array, we can use appropriate isolation materials to solve the problem. Regarding the reproducibility of sensor response and temperature and humidity compensation, they can be solved by packaging methods. False positives can be solved by filtering in the back-end circuit.

### 5.2. Integration with Semi-Flexible Circuits

With the rapid development of flexible processing technologies, wearable sensing devices have achieved remarkable results and are at the forefront of personalized medicine. However, back in 2003, Jung et al. [260] proposed a packaging and interconnect technology for deep textile electronics integration, including a silicon-based micromechanical thermoelectric generator chip for harvesting energy from body heat, opening the way for the development of flexible back-end circuits. With disease-specific smart electronic devices, both patients and clinicians have access to a steady stream of physiological and pathological signals to track treatment progress in real time. To further improve the overall comfort of wearable flexible devices, such as portability, tactile softness, breathability, and biocompatibility, researchers have gradually softened the back-end circuit units. For example, Peng et al. [143] developed a flexible, fully integrated wireless pressure sensing chip system by integrating a sensitivity-tunable sensor with a flexible printed circuit board on a PI film (Figure 11E), which can be applied to different scenarios such as fruit growth, neck pronation, pulse beat, etc., and because of its portability and versatility characteristics, it is in the next generation of wearable devices. In addition, Su et al. [257] created an ion pressure sensing array with optical mode sensitivity and interconnected it with a microcontroller with computational power and an integrated wireless communication module with a flexible back-end circuit, as shown in Figure 11F. Real-time transmission of pressure data to a wearable display was achieved. It meets the requirement of high-precision pressure mapping over a wide pressure detection range and has a broad application prospect in exercise training and rehabilitation. Moreover, similar to the serpentine structure of the sensing array electrodes, Mario et al. [261] proposed a horseshoe-shaped structure while using a narrow metallization scheme to reduce damage to the metal wires, obtaining excellent performance with a bent shape and multiple conductor wires to allow the structure of highly deformed electronic circuits.

For many years, ECGs, which measure the electrical activity of the heart, have been a routine part of clinical monitoring of a patient’s cardiovascular status. ECG devices have been integrated into many commercially available wearable devices to enable individuals to monitor cardiac activity in their daily lives. However, the portability of ECG devices can be somewhat limited due to their weight and bulkiness, which have been routine parts of clinical monitoring of a patient’s cardiovascular status. ECG devices have been integrated into many commercially available wearable devices to enable individuals to monitor cardiac activity in their daily lives. However, the portability of ECG devices can be somewhat limited due to their weight and bulkiness. Therefore, to address this issue, Hisar et al. [256] demonstrated a non-contact wearable patch integrated with a 28-m-thick piezoelectric polymer PVDF membrane and back-end circuitry, as detailed in Figure 11D. NFC technology was used to power the patch and wirelessly record SCG data, enabling power supply and communication between the wearable patch and a reader (e.g., a smartphone). In the future, if such SCG electrodes are integrated into a multifunctional wearable patch system, a sensor fusion system for continuous cardiovascular monitoring is expected to be realized. In addition, Zhang et al. [262] developed a simple, low-cost, battery-free, multiparameter passive wireless flexible sensor (MPWFS) with a sensing array combined with a mobile terminal to provide features including tracking changes in supine posture, warning of pressure sores, and monitoring falls from bed. The research provides a new approach to the care of long-term bedridden patients and offers a wide range of opportunities for advanced medicine. In conclusion, flexible electronics, through the hybrid integration of sensing arrays and back-end circuits to extend mechanical characteristics, research on the compatibility of new materials and structures with the human body, improving the accuracy and reliability of sensing arrays, and realizing real-time analysis and data transmission, will put the field at the forefront of new products and applications with great potential for personalized medicine [263].

### 5.3. Integration with Fully Flexible Circuits

Although semi-flexible circuits have been widely used in flexible electronic products such as skin gauges, stretchable physiological monitoring devices, and supercapacitors, semi-flexible circuits still use traditional rigid conductive materials (e.g., Cu, Ag, and ITO), which are prone to cracking due to metal fatigue or a small bending radius during long-term use, resulting in degradation of electrical performance. Therefore, the development of fully flexible back-end circuits is an inevitable development direction for flexible sensor devices. Zheng et al. [264] successfully prepared core-shell-structured silver liquid metal particles (Ag@LMPs) with excellent electrical conductivity by coating nano-silver on the surface of LMPs through in situ chemical reduction. Fully flexible circuits using Ag@LMPs by screen printing show excellent durability and electrical self-healing due to their functional core-shell structure, which has a promising future in the fabrication of fully flexible circuits. Wang et al. [258] used a stencil printing method to pattern liquid metals on TPU films. The flexible circuits, resistors, capacitors, inductors, and their composite devices were then parametrically prepared by a layer-by-layer assembly technique, as shown in Figure 11G. The resulting devices have perfect stretchability, permeability, and stability. This strategy allows the flexible production of various flexible composite circuits. It is of epoch-making significance for the realization of fully flexible back-end circuits. In addition, Long et al. [259] proposed a new flexible carbon nanotube-based thin film transistor (CNT-TFT) material device (Figure 11H). Replacing the conventional rigid conducting material device, a high-speed and low-power fully flexible circuit based on flexible CNT-TFTs is demonstrated. This new fully flexible circuit may be promising to provide a completely new idea for flexible wireless systems for next-generation healthcare applications.

## 6. Conclusions and Outlook

The development of flexible wearable pressure sensing arrays has reached an unprecedented level in the past few years due to the rapid growth in demand for flexible wearable devices and the emergence of various new materials/structures. This paper reviews the recent research progress on flexible pressure sensing arrays applied to the skin’s surface. A comprehensive description is given in terms of materials, structures, and preparation methods. These factors influence the performance associated with the devices, including extraordinary sensitivity, a wide operating range, stable sensing, a fast response time, low hysteresis, and linearity sensing. Novel materials and microstructures with unique sensing properties are analyzed in detail, and their preparation processes are compared. In addition, important applications in healthcare, human-machine interaction, and other fields are discussed. Finally, the development status of back-end circuits and the connection between flexible sensing arrays and back-end circuit parts are discussed.

Many remarkable achievements have been made in achieving extraordinary performance (e.g., ultra-high sensitivity, wide detection range, linearity, etc.) and comfort (e.g., water resistance, breathability, etc.) of the arrays, but there are still some challenges as well as future opportunities to be addressed in realizing practical applications of epidermally flexible pressure sensing arrays. (1) An unavoidable problem in the fabrication of sensing arrays is the crosstalk between adjacent sensor pixels. The distance between adjacent pixels of the sensing array is the main factor that causes crosstalk in the array. In the future, researchers should overcome the crosstalk problem by conducting extensive experiments to determine the appropriate distance of sensing units. (2) In order to monitor human health in real time and for applications in artificial intelligence systems, a lot of multidisciplinary research is needed to develop wearable sensing systems that integrate multifunctional sensors (e.g., temperature, pressure, and physiological signals) with some additional properties (electromagnetic shielding, self-healing, sensing and actuation functions, and power systems) and real-time data wireless transmission modules. (3) Some special biomedical or artificial intelligence applications require high-density, low-cost pressure sensing arrays to efficiently collect signals from various components, but there is a lack of low-cost, high-precision methods for preparing pressure sensors for large areas. Work in these areas will be an important step in bringing flexible and wearable pressure sensors into future practical applications.

In summary, flexible pressure sensing arrays are proposed based on the human skin sensing mechanism, just as human skin is a complete system that starts by sensing stimuli and then transmits the signal to the brain, which makes the corresponding commands. Future skin-based wearable devices should not just be a single extraordinary-performance sensing array that can be realized but should be combined with signal processing, PC end, and IOT technology to form a whole system so that it can transition from a laboratory device to a device that can solve people’s daily life problems. Only by solving these limitations of sensing arrays and combining them with advanced electronic technologies to really enter into daily life applications can we really take a big step forward in the Internet era.

## Figures and Tables

**Figure 1 biosensors-13-00656-f001:**
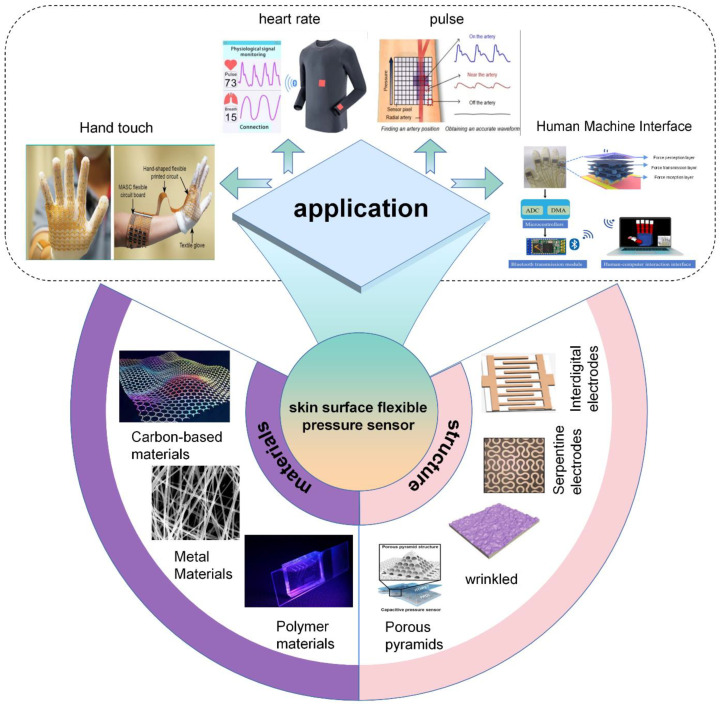
Review of materials, structures, processes, performance, and application of flexible pressure sensing arrays.

**Figure 2 biosensors-13-00656-f002:**
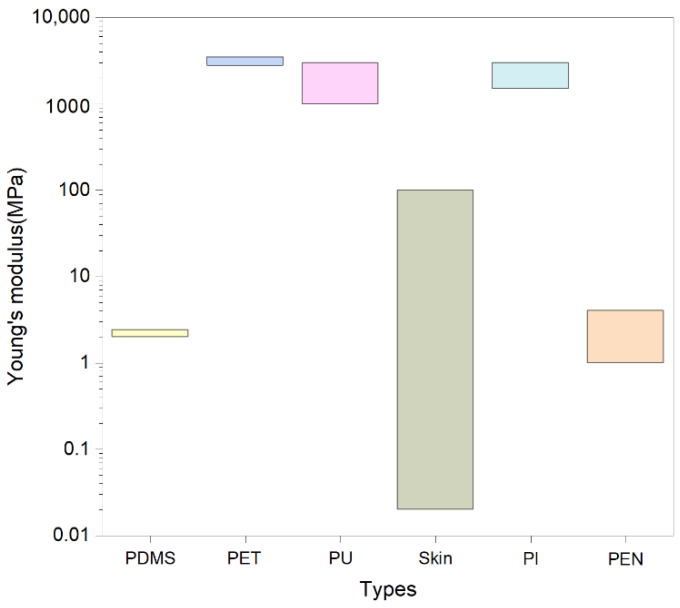
Comparison of different materials and skin moduli.

**Figure 3 biosensors-13-00656-f003:**
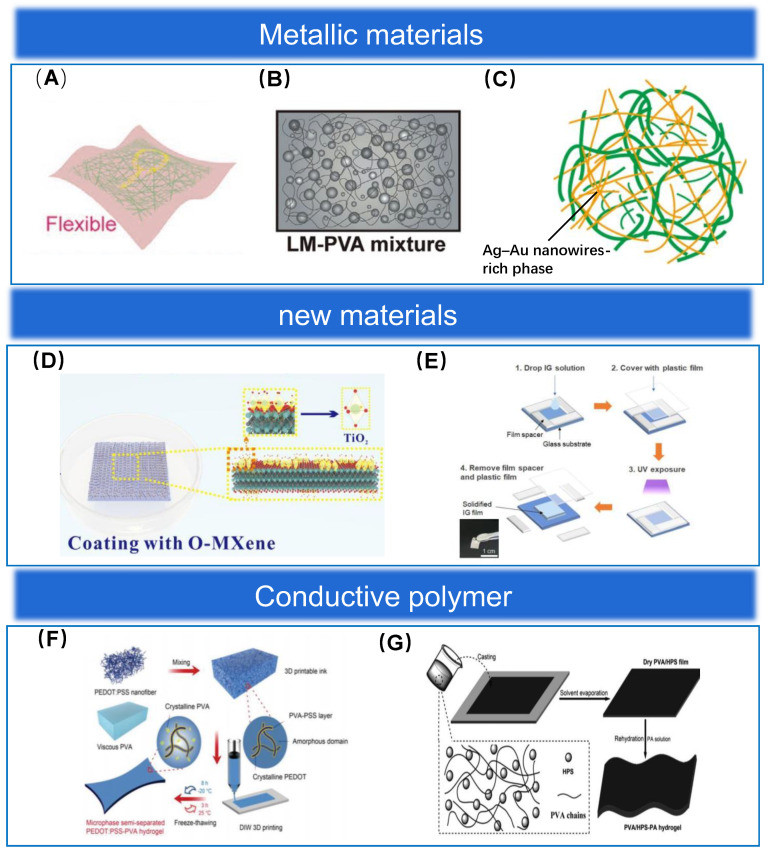
(**A**) gold nanoflowers (AuNFs) schematic diagram. Reprinted with permission from John Wiley and Sons [53]. (**B**) Liquid metal−polyvinyl alcohol (LM−PVA) mixture. Reprinted with permission from Elsevier [21]. (**C**) Silver (Ag) and gold (Au) nanowire composites. Reprinted with permission from Springer Nature [54]. (**D**) Coating with O−MXene. Reprinted with permission from John Wiley and Sons [55]. (**E**) Fabrication method of ionic−gel (IG) membrane. Reprinted with permission from Elsevier [56]. (**F**) Principle and fabrication process of the PEDOT:PSS−PVA conducting polymer hydrogel. Reprinted with permission from John Wiley and Sons [57]. (**G**) Schematical illustration of the fabrication process for composite hydrogels. Reprinted with permission from Elsevier [58].

**Figure 4 biosensors-13-00656-f004:**
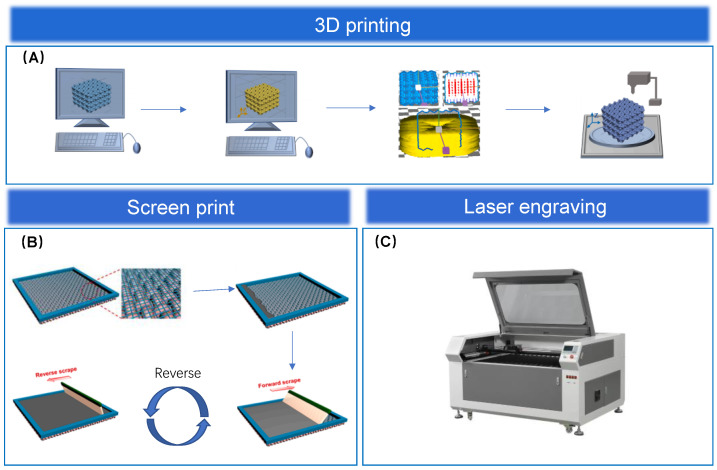
(**A**) Schematic illustration of the 3D printing process. Reprinted with permission from Elsevier [93]. (**B**) Schematic illustration of the screen-printing process. Reprinted with permission from Elsevier [94,95,96]. (**C**) Laser engraving machine.

**Figure 5 biosensors-13-00656-f005:**
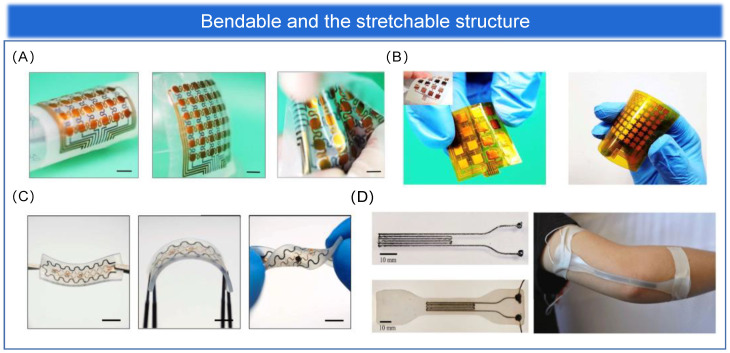
(**A**) The device still retains its original integrity even under mechanical deformation, such as bending, stretching, or twisting. Reprinted with permission from Springer Nature [127,128,129]. (**B**) Optical images of the flexible sensing array at the rest (transferred electrode (insert)) and bent states, respectively. Reprinted with permission from IEEE [127,128,129]. (**C**) Images of the integrated sensor patch (ISP) in elastic compliance under no load (left), bending (middle), and torsion (right). Reprinted with permission from Elsevier [127,128,129]. (**D**) Liquid metal alloy eutectic gallium indium (EGaIn) printed and encapsulated to perform as a strain sensor to detect elbow flexion and extension. Reprinted with permission from the American Association for the Advancement of Science [133].

**Figure 6 biosensors-13-00656-f006:**
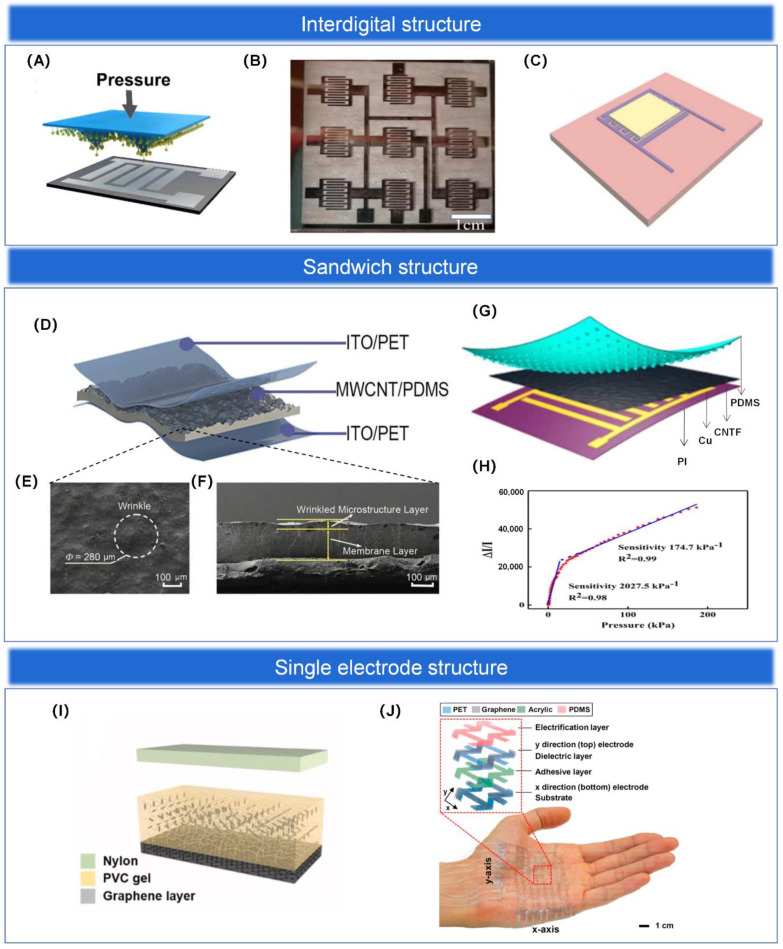
(**A**) Schematic showing the sensor structure. The gold nanowires/PDMS (AuNWs/PDMS) film comes into contact with the printed interdigitated electrodes on the PEN films. The contact area between the two electrodes will increase with increased applied pressure because of the deformation of pyramid microarrays. Reprinted with permission from the American Chemical Society [145]. (**B**) Interdigital electrode and circuit of the 3 × 3 sensing array. Reprinted with permission from Springer Nature [146]. (**C**) Schematic diagram of the fully sprayed MXene−based and interdigital electrode extraordinary performance pressure sensor. Reprinted with permission from John Wiley and Sons [119]. (**D**) Schematic of the capacitive sensor and photograph of the sensor. Reprinted with permission from John Wiley and Sons [35]. (**E**) SEM photograph of the surface of the multi−walled carbon nanotubes/PDMS (MWCNTs/PDMS) dielectric layer. Reprinted with permission from John Wiley and Sons [35]. (**F**) Cross−section SEM image of the dielectric layer. Reprinted with permission from John Wiley and Sons [35]. (**G**) Schematic diagram of the hierarchy of flexible pressure sensors. Reprinted with permission from American Chemical Society [147]. (**H**) Sensitivity measurement results of the sensors. Reprinted with permission from American Chemical Society [147]. (**I**) Structure illustration of the polyvinyl chloride (PVC) gel and multi−layered graphene−based stretchable and biocompatible single electrode type triboelectric nanogenerator (PG−TENG). Reprinted with permission from Elsevier [148]. (**J**) Optical image and scheme of self−powered single electrode−TENG (SE−TENG) touch sensor with auxetic design. The device consists of top and bottom graphene electrodes, a PDMS electrification layer, and a PET substrate. Reprinted with permission from Elsevier [149].

**Figure 7 biosensors-13-00656-f007:**
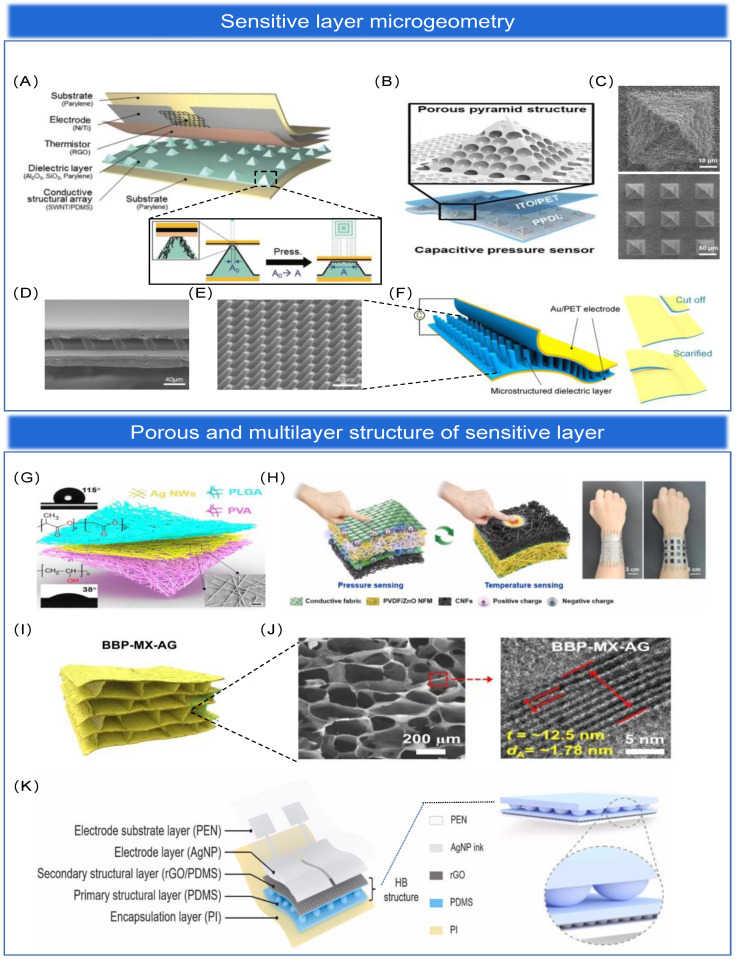
(**A**) Schematic illustration of stimulus−discriminating and linearly sensitive bimodal electronic skin (E−skin) and SEM images of a spray−coated rGO thermistor. Reprinted with permission from John Wiley and Sons [152]. (**B**) Structure diagram of a porous pyramid dielectric layer (PPDL). Reprinted with permission from the American Chemical Society [138]. (**C**) The top view and scanning electron microscopy SEM images of the PPDL. Reprinted with permission from the American Chemical Society [138]. (**D**,**E**) Photographs and SEM images showing the highly uniform micropillar array and flexible and ultrathin device. Reprinted with permission from the American Chemical Society [155]. (**F**) The sensor architecture. Mechanical damages such as cuts or scratches do not affect the sensing performance. Reprinted with permission from the American Chemical Society [155]. (**G**) Schematic illustration of the three−dimensional network structure of the all−nanofiber TENG−based e−skin. The images of the water contact angle and molecular structure of poly(lactic−co−glycolic acid (PLGA) and PVA are inserted on the top left and lower left, respectively. The surface SEM image of the AgNWs electrode is inserted on the lower right. Reprinted with permission from the American Association for the Advancement of Science [159]. (**H**) Schematic diagram of dual−model textile for pressure and temperature sensing and optical photographs of the front side and reverse side of electronic skin textile (4 × 4 pixels) conformably attached on an arm. Reprinted with permission from Elsevier [180]. (**I**) Illustration of the structure of Bottle Brush PDMS−MXene−aerogel (BBP−MX−AG). Reprinted with permission from Springer Nature [160]. (**J**) SEM image of BBP-MX-AG and images showing nanochannels with an interlayer spacing of 1.8 nm in the cellular wall of BBP−MX−AG. Reprinted with permission from Springer Nature [160]. (**K**) Schematic diagram of the hierarchical branching (HB) sensor, which contains two hemisphere−structured layers and a side−by−side electrode layer. The reduced graphene oxide (rGO) sheets coated on the secondary layer form a wrinkled morphology on the surface of elastic PDMS, serving as the tertiary layer. Reprinted with permission from Elsevier [181].

**Figure 8 biosensors-13-00656-f008:**
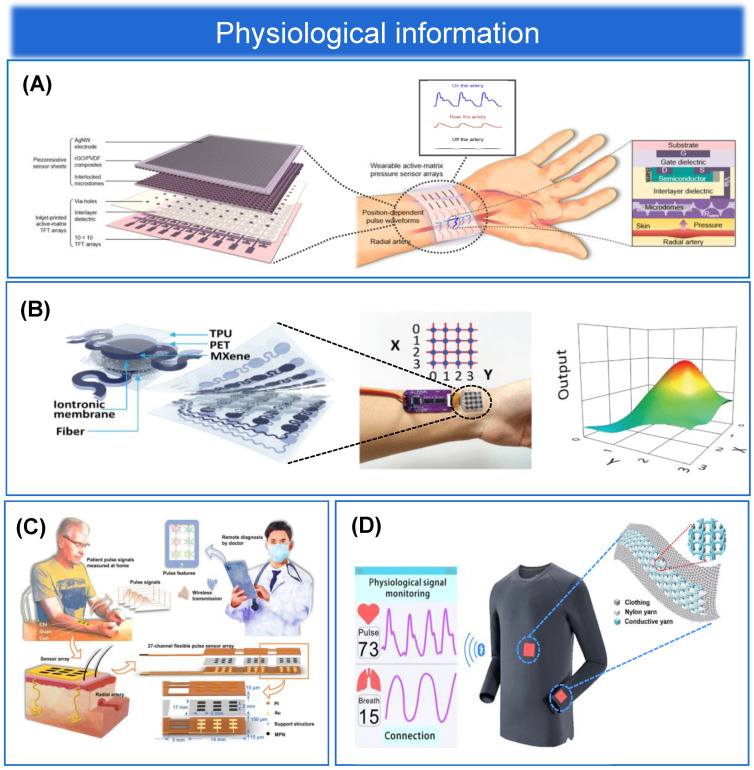
(**A**) Design of the wearable active−matrix pressure sensing arrays. Reprinted with permission from the American Chemical Society [204]. (**B**) The coordinates of the units in the sensing array attached to the wrist of the volunteer and a schematic diagram of a stretchable inotropic pressure sensor (SIPS). Reprinted with permission from IEEE [205]. (**C**) Schematic diagram of an intelligent disease diagnosis system for pulsed signals. Reprinted with permission from the American Chemical Society [206]. (**D**) Two triboelectric all−textile sensing arrays (TATSAs) integrated into a shirt for the monitoring of pulse and respiratory signals in real time. Reprinted with permission from the American Association for the Advancement of Science [207].

**Figure 9 biosensors-13-00656-f009:**
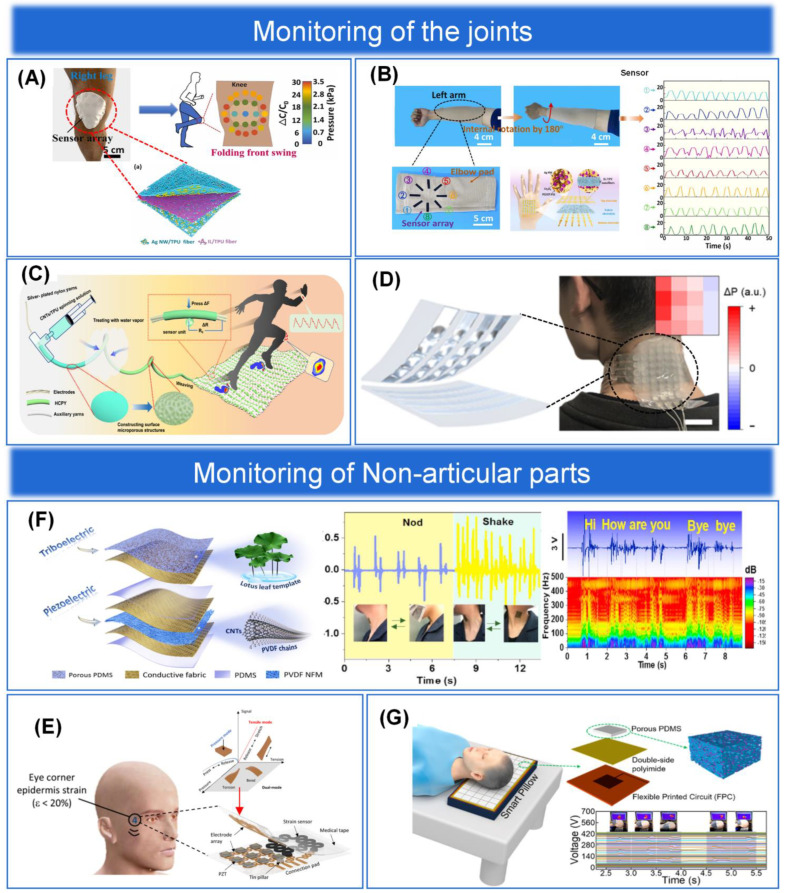
(**A**) Flexible, stretchable, breathable, sweat−resistant all−nanofiber ionic electrons Haptic sensing array for continuous and comfortable knee movement monitoring. Reprinted with permission from Elsevier [216]. (**B**) Wearable, ultrathin, and breathable tactile sensors with an integrated all−nanofiber network structure for highly sensitive and reliable motion monitoring. Reprinted with permission from Elsevier [217]. (**C**) Hierarchical core−shell piezoresistive yarns (HCPYs) manufacturing schematic diagram and spatial pressure distribution map of human joints. Reprinted with permission from the American Chemical Society [218]. (**D**) A photograph showing the sensing array attached to the cervical region of the volunteer. Reprinted with permission from the American Chemical Society [219]. (**E**) Detection capabilities for human activities of subtle physiological movements; an explosive schematic illustration of the tensile strain−pressure sensor. Reprinted with permission from Elsevier [129]. (**F**) Energy autonomous hybrid electronic skin structure with multi−modal sensing capability and its application. Reprinted with permission from Elsevier [220]. (**G**) Application of flexible and breathable TENG (FB−TENG) array as a smart pillow for monitoring head movement. Reprinted with permission from the American Chemical Society [221].

**Figure 10 biosensors-13-00656-f010:**
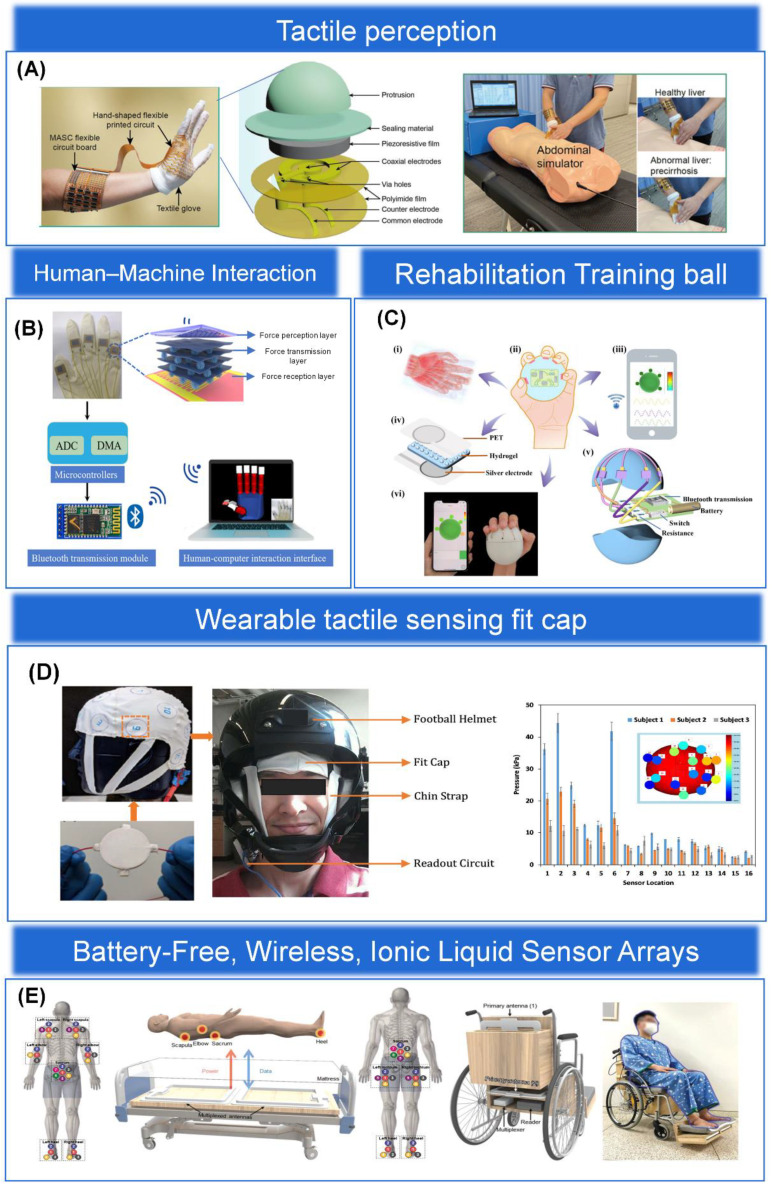
(**A**) Structure and application of a scalable tactile glove. Reprinted with permission from John Wiley and Sons [239]. (**B**) Schematic diagram of the wearable gesture recognition device system. Reprinted with permission from the American Chemical Society [214]. (**C**) Demonstrations of the piezoresistive sensor in rehabilitation training. Reprinted with permission from the American Chemical Society [240]. (**D**) Wearable tactile sensing fit cap demonstration. Reprinted with permission from the American Chemical Society [241]. (**E**) Battery−free, wireless, ionic liquid sensing arrays to monitor the pressure and temperature of patients in beds and wheelchairs. Reprinted with permission from John Wiley and Sons [242].

**Figure 11 biosensors-13-00656-f011:**
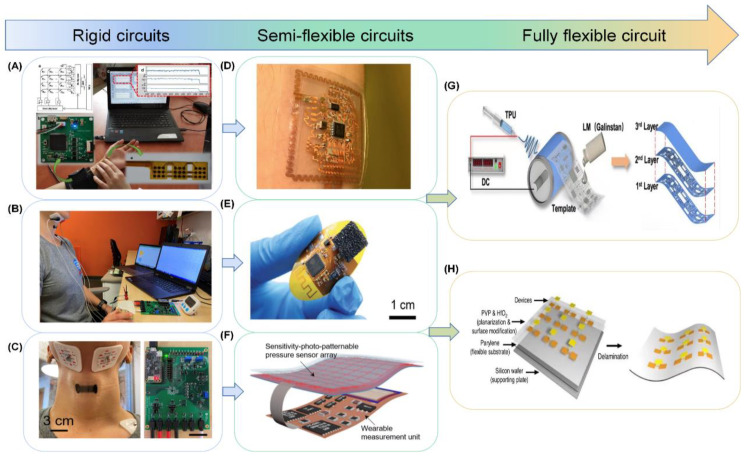
(**A**) The schematic diagram of the array scanning circuit system, an optical image of the array scanning circuit system, and an optical image of the 27−channel flexible pulse perception array. Reprinted with permission from the American Chemical Society [206]. (**B**) Close−up image of the strain and EMG sensors on the throat of the participant and the MAX30001 evaluation kit used in the experiments. Reprinted with permission from the American Chemical Society [254]. (**C**) A photograph shows the testing setup where the sensors are connected to the evaluation kit, which is connected to a laptop, and the participant is seated in a relaxed position. Reprinted with permission from the American Chemical Society [254]. (**D**) Top view of the three−layer system−in−a−foil patch to measure seismocardiography (SCG) signals from a person’s torso. The final dimension of the patch is 45 mm × 55 mm. Reprinted with permission from the American Chemical Society [256]. (**E**) Photographic image of the sensing chip system. Reprinted with permission from the Royal Society of Chemistry [143]. (**F**) Schematic illustration of the wearable system consisting of a sensitivity−photo−patternable pressure sensing array and a wearable measurement unit. Reprinted with permission from the American Chemical Society [257]. (**G**) Design schematic of the flexible electronic system, including the TPU substrate prepared by electrospinning and the LM circuit patterned on the TPU membrane by stencil printing. an electronic system with a multilayer structure can be prepared through layer−by−layer assembly. Reprinted with permission from the American Chemical Society [258]. (**H**) Schematic illustration of device and circuit fabrication on a flexible parylene substrate. Reprinted with permission from Springer Nature [259].

**Table 1 biosensors-13-00656-t001:** Mechanical and electrical properties of different materials.

Material	Breaking Elongation	Tensile Strength	Compressive Strength	Elastic Modulus	Resistivity	Refs
Gold nanofibers	150%	3.4–14 GPa	1.5–8 GPa	65–168 GPa	3.9 ± 0.4 μΩ·cm	[77,78,79,80]
LM−PVA (18%LM)	540 ± 36%	16 ± 3 MPa	—	217 ± 46 MPa	1.2 MPa^−1^ (35%LM)	[21]
Au−Ag	840%	—	—	3.28 MPa	41,850 s cm^−1^	[54]
Graphene	About 320%	130 ± 10 GPa	—	1 ± 0.1 TPa	200 Ω·m ^−1^	[81]
MXenes	—	—	—	0.33 ± 0.3 TPa	2–1500 s cm ^−1^	[70,82]
PEDOT:PSS	82 ± 1%	—	—	6.8 MPa	1010 ± 5 s cm ^−1^	[83]
Ionic gel	1500–2500%	2.5–16 MPa	—	0.2–18 MPa	2.25 s m^−1^	[84]

**Table 2 biosensors-13-00656-t002:** Comparison table of various printing technical parameters.

	Range of the Dimension	Operating Temperature	Printing Time	The Minimum Feature Size	Refs
DLP	100 × 60 mm–190 × 120 mm	Room temperature	25–100 mm/min	50 μm	[27,97]
SLA	120 × 68 × 155 mm	Room temperature	—	Horizontal: 47 μm Vertical: 10 μm	[75]
LCD	115 × 65 × 165 mm	—	20 mm/h	Horizontal: 47 μm Vertical: 1.25 μm	[98]
FDM	Single nozzle: 305 × 305 × 300 mm double nozzle: 280 × 305 × 300 mm	15–30 °C	30–150 mm/s	10 μm	[99]
DIW	—	—	$150\;{mm}^{3}/s$	500 μm	[100]
IJP	—	Room temperature	—	10 μm	[101]

**Table 3 biosensors-13-00656-t003:** Structures of common electrodes and sensitive layers.

Type	Remark	Polymer Matrix	Key Material	Sensitivity [kPa^−1^]	Response Time [ms]	Detection Range [kPa]	Refs.
Interdigital electrodes	Interdigital electrodes+ micro−pyramid	PEN, PDMS	AuNWs/PDMS, Silver nanoparticles (AgNPs)	23	<10	<600 Pa	[145]
	Interdigital electrodes+ wrinkled	PI, PU	MXene/PU, MXene	509.8	67.3	—	[119]
Serpentine electrodes		PVA, Si wafer	Ti/Platinum (Pt)	—	—	—	[150]
		PI	PI, Ag/AgCl	—	—	—	[134]
Micropyramid		PDMS	PDMS/CNTs, chromium (Cr)/Au	0.16 or 0.04	48	<1 or 0.75–2.5	[151]
	Wrinkled micro−pyramid+ interdigital electrodes	parylene	Nickel (Ni)/Ti, reduced graphene oxide (rGO), Single−walled Carbon Nanotubes (SWCNT)/PDMS	0.7	50	0–25	[152]
	Porous micro−pyramid	Ecoflex, PDMS	Indium tin oxide (ITO)/PET, PDMS	44.5	—	<100 Pa	[138]
		Ecoflex, PDMS	PDMS, AgNWs/PDMS	449	9	0.14 Pa	
Micropillar/needle		PDMS	Ag/Ni, PDMS	—	50	0–145	[153]
		PI, PDMS	Au/PDMS, Au/PI	33.16	9	12–176	[154]
		PET	PDMS, Au/PET	0.42 or 0.04	<70	<1.5 or >5	[155]
Microdoe	Microdome + lamellar	PET	rGO/PVDF, Ag/copper (Cu)	47.7	20	0.0013–353	[156]
		PDMS	PVDF, PDMS/Au	30.2	25	0–130 Pa	[140]
Porous	Foam	—	Zinc oxide nanoparticles (ZnONPs)/MWCNTs/PDMS	—	—	1–25	[157]
	Sponge	PI	molybdenum disulfide/hydroxyethyl cellulose/PU (MoS_2_/HEC/PU), Ag	0.746	120	250	[158]
	Nanofiber	—	AgNWs, poly(lactic−co−glycolic acid (PLGA), PVA	0.011	—	—	[159]
	Aerogel	—	MXene/PDMS	>1900	—	—	[160]
Wrinkled	Sandpaper	PI	PVA, Au	>220	<10	0.08 Pa–360 kPa	[161]
		PET	MWCNTs/ PDMS, ITO/MXene/PET	1.448	—	0.005–21	[35]
Bionic	Dendritic + lamellar	natural rubber	SWCNT/ PVP/	165.4	48	—	[162]
	Multiscale hierarchical + rose petal	PDMS	PDMS	120 (<0.5 kPa Press)	30	0.88 Pa–32 kPa	[163]

## Data Availability

The data presented in this study are available on request from the corresponding author. The data are not publicly available due to privacy issues.

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
