# Peer review of "A Review of Epidermal Flexible Pressure Sensing Arrays"

_biosensors, 2023, doi:10.3390/bios13060656_

Round 1

Reviewer 1 Report

This review (biosensors-2431960) focuses on flexible pressure sensing arrays and their application. This topic is a hot research field and has a wide range of readers, including in the fields of pressure sensors, flexible electronics, and various applications. In all, this review is well written and organized. But there are many problems to be solved before possible publication. My specific comments are as follows:

1.     Figure 1 and the following discussions: The application is not fully summarized. For example, the application of sensors in human-computer interaction is missing.

2.     Table 1: The authors should provide references.

3.     Figure 4 (material fabrication process): Ink jet printing (belonging to 2D printing) is also a common process for preparing pressure sensors. However, the authors did not review the relevant content.

4.     2.1.2 Electrode Materials: Flexible polyester fiber conductive tape can be directly used to prepare flexible pressure sensors. Suggest adding relevant discussions.

5.     2.2 and 2.3 electrodes and sensing/conductive active materials should not miss the eco-friendly and low-cost amorphous daily carbon ink (J. Mater. Chem. C, 2023,11, 5585-5600).

6.     3.3.1 Microgeometric structure: “flexible substrates (microstructured PDMS, PU, Ecoflex, silicon flakes, etc.).”. Flexible paper with natural microstructure, large area, eco-friendly properties should not be ignored. Flexible paper has been widely reported for the preparation of flexible pressure (resistance, capacitance) sensors and pressure sensor arrays.

7.     Table 2: What is the basis for the authors’ selection of literature? The authors have not completely summarized the latest research progress in this field.

8.     Figure 8: Blood pressure monitoring is also a research hotspot in the field of pressure sensors. However, the authors did not review the relevant content.

9.     Pressure sensors usually include resistive, capacitive, piezoelectric, and triboelectric types. The authors should introduce the relevant content separately.

10.  Sensing arrays: Signal crosstalk usually occurs in sensing arrays. It is suggested that the authors use an independent section to summarize the relevant contents.

11.  Figures’ quality needs to be improved, such as Figure 2.

12.  Check the format/style of the target journal. Such as reference. Complete journal volume number, page/literature number, and year. The journal name needs to be abbreviated.

13.  English writing of the manuscript needs to be polished.

English writing of the manuscript needs to be polished

Reviewer 2 Report

The review focus its investigation on epidermal flexible pressure sensing arrays claiming to cover not only the materials and manufacturing of the devices but also applications and related back-end electronics. The paper has an extensive part that deals with materials and applications but lacks of information regarding flexible related electronics. More precisely, the novelty of this paper should relies on back-end electronics, providing a whole scenario of different typologies of flexible and semi-flexible electronics to be integrated in these wearables arrays. Regarding this specific aspect, I strongly encourage authors to strengthen this part of the paper focusing also on specific circuit arrays, methods of connection between flexible and rigid part of the PCB, strategies to treat sensor signals and a section regarding sensor noise and input digitalization.

Additionally a point-by-point list to be consider is reporting below:

Introduction

a)The sentence related to lines 75-80 are not very clear. I suggest to rephrase them for a better readability. It is not clear the differences with the previous reviews. So I strongly suggest to improve this crucial part, trying to point out the importance of presenting also back-end circuitry as part of the sensing technology.

b) In my opinion, in the introduction, all the most important active materials have to be listed to better resume for the readers what they will find in the review. Please cite among the polymeric materials PVDF and its copolymers (PVDF-TrFE) referring this class of polymer based devices with the following papers

1)      R. S. Dahiya, G. Metta, M. Valle and G. Sandini, "Tactile Sensing—From Humans to Humanoids," in IEEE Transactions on Robotics, vol. 26, no. 1, pp. 1-20, Feb. 2010, doi: 10.1109/TRO.2009.2033627.

2)      F. Maita et al., "Ultraflexible Tactile Piezoelectric Sensor Based on Low-Temperature Polycrystalline Silicon Thin-Film Transistor Technology," in IEEE Sensors Journal, vol. 15, no. 7, pp. 3819-3826, July 2015, doi: 10.1109/JSEN.2015.2399531.

3)      Viola, F.A., Spanu, A., Ricci, P.C. et al. Ultrathin, flexible and multimodal tactile sensors based on organic field-effect transistors. Sci Rep 8, 8073 (2018). https://doi.org/10.1038/s41598-018-26263-1

c) The introduction is rather short. I suggest to the authors to motivate their choice of focusing on the read out and the back-end circuits of pressure sensors with more arguments related to the advantages in reducing power consumptions of the devices, size and dimensions, response time, sensor cost. Additionally, here authors should highlight if they focus the review only to skin surface devices or wearables more in general and which kind of contact with the skin is investigated (i.e. issues related to sweat, adhesion on the skin, breathability of the devices, sensor durability, etc.). Moreover, it should be pointed out something regarding the sustainability of the manufacturing of these sensors and circuitry. Which is the environmental impact of the fabrication processes, there are some biodegradable pressure sensors? Are the sensors disposable or not? In my opinion these topic should be mentioned.

2.1.2 Electrode Materials

a) Please be sure to report the complete name of a material before mentioning its acronym (for example LM-PVA full name is reported only in the caption of the figure 3 and not in the text).

b) Please report information about biocompatibility or potential skin sensibilization or allergies of materials like MXene or liquid metals (gallium based materials).

2.3 Sensitive Layer Materials

a) Line 268. Please insert some numbers regarding hydrogel based devices stability and durability (avoid terms like good or not good).

b) line 276 Please explain in which configuration is used the PANI (bulk, paste, nanofiber, micromesh, etc.)

c) It not clear why this paragraph is so small and it only refers to few active materials (specifically only, hydrogels, hydrogel blend and PANI). Also the properties of the other materials mentioned in the introduction should be reported.

d) I strongly suggest to insert in this paragraph the physical working principles of the active material for fabricating the pressure sensors (i.e. change in the dimensions, piezoelectric device, triboelectric device, etc.). Additionally, in the next paragraph they report microstructuring and foam of active layers. I strongly suggest to clarify the physical need to use such structures to enhance the sensitivity. Authors tend to mix the feature of a material with the sensor and this is detrimental for the comprehension of the review. I suggest to divide the type and the properties of active materials with the sensors themselves to improve the clarity of the text.

2.4 Material Fabrication Process

a) In this paragraph a table with “the range of the dimension of the printed object”, “operating temperature”, “printing time” and “the minimum feature size” of the should be reported to visualize immediately the advantages and the limitations of each technology.

b) line 370-371, line 390, line 407-408, line 430, please report numbers (again avoid to use the words good, high resolution)

c) in 2.4.4 others authors should consider also lamination methods, micro and nano-imprinting lithography as fabrication method.

3.1.1 Bendable structure

a) please report a mathematical section in which authors can explain the dependence of the flexibility of a device from the thickness of the sensor substrate as  mentioned in H. Gleskova, S. Wagner, Z. Suo Appl. Phys. Lett. 75, 3011–3013 (1999). this is very useful for understading mechanical and bending stress of a device depoyed on the human skin

b) please cite these articles related to ultra-thin and super bendable devices:

-          Sunghoon Lee et al., Nanomesh pressure sensor for monitoring finger manipulation without sensory interference.Science370,966-970(2020).DOI:10.1126/science.abc9735

-          Kaltenbrunner, M., Sekitani, T., Reeder, J. et al. An ultra-lightweight design for imperceptible plastic electronics. Nature 499, 458–463 (2013). https://doi.org/10.1038/nature12314

3.3.2 Porous structure

a) At the line 805  the term electronic skin is introduced. Please explain a little bit what scholars intend with this terminology.

3.4 Packaging method

This paragraph should be rewritten with a deeper analysis of the different packaging methods and how they influences not only the encapsulation and insulation from the environment but also regarding the changes in sensor features. As is the section is too general and useless for a reader.

4.1 Physiological Information Monitoring

a) Please substitute the term “Traditional Chinese Medicine (TCM)” with the word medicine. The non- traditional medicine is not a science, so it is meaningless.

4.2.1 Joint Areas

a) Please add the following papers that consider wristband-based devices or wearable systems to detect biophysical parameters or for motion recognitions:

-          T. Chouhan, A. Panse, A. K. Voona and S. M. Sameer, "Smart glove with gesture recognition ability for the hearing and speech impaired," 2014 IEEE Global Humanitarian Technology Conference - South Asia Satellite (GHTC-SAS), Trivandrum, India, 2014, pp. 105-110, doi: 10.1109/GHTC-SAS.2014.6967567.

-          A. Ferrone et al., "Wearable band for hand gesture recognition based on strain sensors," 2016 6th IEEE International Conference on Biomedical Robotics and Biomechatronics (BioRob), Singapore, 2016, pp. 1319-1322, doi: 10.1109/BIOROB.2016.7523814.

-          F. Lorussi, Enzo Pasquale Scilingo, M. Tesconi, A. Tognetti and D. De Rossi, "Strain sensing fabric for hand posture and gesture monitoring," in IEEE Transactions on Information Technology in Biomedicine, vol. 9, no. 3, pp. 372-381, Sept. 2005, doi: 10.1109/TITB.2005.854510.

b) authors should mention differences in wearable devices that need an intimate contact with the skin and high sensitive sensors where the devices can be deployed on the garments without affecting the user comfort.

c) line 1122, breathability can be achieved also tailoring the sensor arrays with web like shape (i.e. O. Yue et al., Spider-Web and Ant-Tentacle Doubly Bio-Inspired Multifunctional Self-Powered Electronic Skin with Hierarchical Nanostructure, Adv. Sci.2021,8, 2004377, DOI: 10.1002/advs.202004377), I encourage authors to comment also this statement.

5.1 Integration with conventional rigid circuits

a) Authors should comment in this section the strategies to used commercial or semi-commercial back-end electronics to filter, amplify and digitalize the sensor signals.

b) Authors should add strategies to solve issues related to reproducibility of the sensor response in wearables, cross-talk and false positive. Also temperature and humidity compensation should take into account.

5.2 Integration with semi-flexible circuits

a) regarding the semi-flexible circuits please cite the following papers:

-          S. Jung, C. Lauterbach, M. Strasser and W. Weber, "Enabling technologies for disappearing electronics in smart textiles," 2003 IEEE International Solid-State Circuits Conference, 2003. Digest of Technical Papers. ISSCC., San Francisco, CA, USA, 2003, pp. 386-387 vol.1, doi: 10.1109/ISSCC.2003.1234347.

-          D. Corzo et al., Flexible Electronics: Status, Challenges and Opportunities, Front.Electron., 30 September 2020, Sec. Flexible Electronics, Volume 1 - 2020 | https://doi.org/10.3389/felec.2020.594003

-          M. Gonzalez et al., Design of metal interconnects for stretchable electronic circuits, Microelectronics Reliability, Volume 48, Issue 6, June 2008, Pages 825-832

-          Fabrice Axisa, Jan Vanfleteren, Thomas Vervust, Method for manufacturing a stretchable electronic device, US Patent 8,207,473

Conclusions

This part is too long and it must be completely rewritten, eliminating speculating parts that need to be inserted in the previous sections.

References

Even if the numbers of references is coherent for a review, the majority of the cited papers come from Chinese groups. Authors should take into account the most valuable papers by authors from all the rest of the world, where the research on ultra-thin and conformable pressure sensors is pursued on different sectors. For example

-          T. Ali et al., Screen-printed ferroelectric P(VDF-TrFE)-co-PbTiO3 and P(VDF-TrFE)-co-NaBiTi2O6 nanocomposites for selective temperature and pressure sensing, ACS Appl. Mater. Interfaces 2020, 12, 34, 38614–38625

-          T. Someya et al., A large-area, flexible pressure sensor matrix with organic field-effect transistors for artificial skin applications, PNAS June 28, 2004 101 (27) 9966-9970,  https://doi.org/10.1073/pnas.0401918101

-          E. Hosseini, Glycine–Chitosan-Based Flexible Biodegradable Piezoelectric Pressure Sensor ACS Appl. Mater. Interfaces 2020, 12, 8, 9008–9016, https://doi.org/10.1021/acsami.9b21052

Really minor editing required

Round 2

Reviewer 1 Report

I carefully checked the response and revised manuscript. Concerns of  reviewer have been addressed well and publication is recommended.

Reviewer 2 Report

The Authors have answered to all my points, so I consider the paper ready for the publication.

As minor point, please note that at line 1385 the sentence " The strain sensors are not in direct contact with human skin" is repeated, please amend this part.